# Beyond Fixed Biases: Decoding the Role of Reasoning Uncertainty in MLLM Modality Conflicts

Zhuoran Zhang [* 1 2 3 4]  Tengyue Wang [* 2 4 5]  Xilin Gong [2 6]  Yang Shi [1]  Haotian Wang [7]  Di Wang [2 3]  Lijie Hu [4]

## Abstract

Multimodal Large Language Models (MLLMs) must resolve conflicts when modalities provide contradictory information, a behavior we term "modality following". We propose a framework that decomposes this behavior into case-specific relative preference uncertainty and stable inherent preference. Across diverse MLLMs and benchmarks, the probability of following a modality consistently decreases as its relative preference uncertainty increases, a trend robust to alternative uncertainty indices. This regularity defines a "balance point" where modality preferences are evenly matched, offering a capability-disentangled measure of modality bias. Layer-wise probing further shows that ambiguous cases near the balance point trigger middle-to-late-layer "concept oscillations," where top predictions vacillate between modality-supported answers. Finally, we demonstrate the framework's utility for preference steering through Supervised Fine-Tuning (SFT). We find that data efficiency is governed by preference uncertainty: training on easy samples (where one modality dominates) fails to generalize, whereas targeting the identified "boundary cases" is essential for robust preference alignment and suppressing internal vacillation.

## 1. Introduction

Multimodal large language models (MLLMs) (OpenAI et al., 2024; Team et al., 2023; Wang et al., 2024; Yin et al., 2024; OpenAI et al., 2024) demonstrate powerful capabilities by processing information from various sources. However, a critical challenge arises when these modalities present conflicting information. For example, an image might show a blue car, while the accompanying text describes it as red. In such cases, the MLLM must resolve the conflict, leading to an observable behavior we term **modality following**: the model's final output aligns with the information from one modality over the other. Prior studies (Zhang et al., 2025; Deng et al., 2025) have typically examined this phenomenon using coarse, dataset-level statistic: the ratio of text-following versus vision-following cases on a given set of conflicting inputs. This overlooks a crucial factor: the model's *confidence* in each of its unimodal predictions. Reasoning uncertainty varies significantly across models and cases, yet macro-statistics treat all instances indiscriminately.

To truly understand the modality-following process, we propose that the static, dataset-level following statistics are emergent properties of two distinct underlying factors: (1) the **relative preference uncertainty** between the two modalities on a case-by-case basis, measured under unimodal inputs, which reflects the model's confidence gap between text-only and vision-only processing, and (2) a more stable, **inherent modality preference**, which we define as the model's intrinsic leaning toward one modality when the preference uncertainties from both are perceived as equal. This leads to our central hypothesis: **An MLLM's modality-following behavior is a dynamic process governed by the interplay between the relative preference uncertainty of the conflicting modalities and the model's own inherent preference.** In simpler terms, a model's decision to follow the text depends on whether the text's reasoning advantage (i.e., its low relative uncertainty compared to the image) is significant enough to overcome the model's potential inherent preference for vision.

Our overall analysis process is shown in Figure 1. To test this hypothesis, we utilize a combination of controllable synthetic data and diverse real-world benchmarks. This dual approach allows us to systematically isolate reasoning difficulty in controlled settings while ensuring our findings generalize to authentic, complex scenarios. Across all evaluations, we anchor our analysis on a case-specific measure

*Equal contribution [1]Peking University [2]Provable Responsible AI and Data Analytics (PRADA) Lab [3]King Abdullah University of Science and Technology [4]Mohamed bin Zayed University of Artificial Intelligence (MBZUAI) [5]South China University of Technology [6]University of Georgia [7]Tsinghua University. Correspondence to: Di Wang <di.wang@kaust.edu.sa>, Lijie Hu <lijie.hu@mbzuai.ac.ae>.

*Proceedings of the 43rd International Conference on Machine Learning*, Seoul, South Korea. PMLR 306, 2026. Copyright 2026 by the author(s).

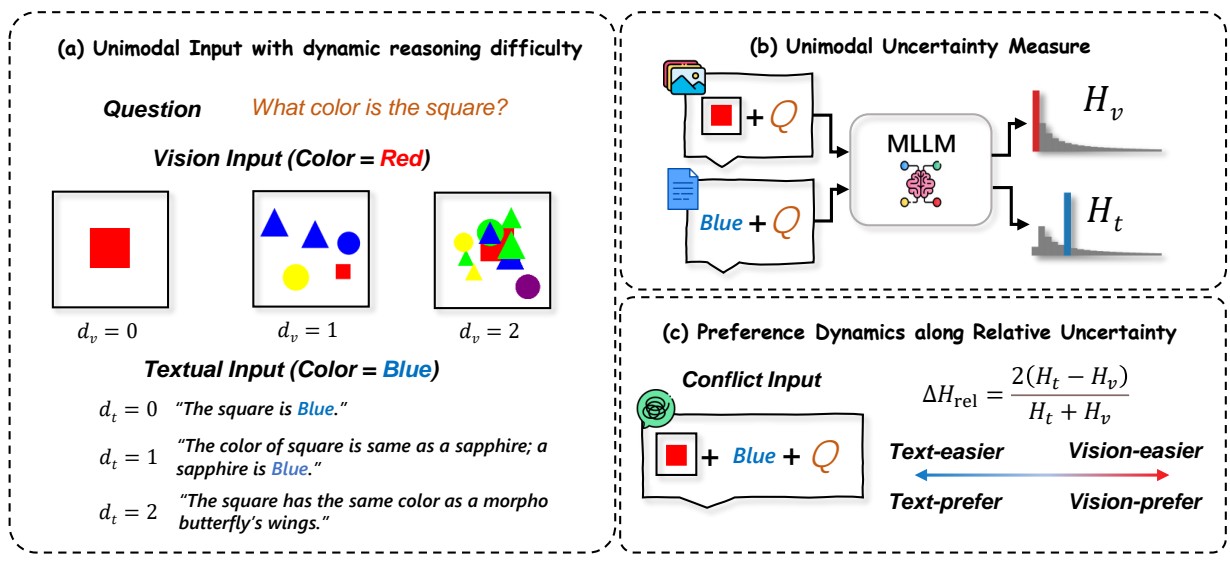

*Figure 1.* Overview of the analytical framework. **(a)** We create inputs with independently controllable visual ($d_v$) and textual ($d_t$) difficulty. **(b)** We measure the model's perceived uncertainty for each modality via output entropy ($H_v$, $H_t$). **(c)** We then use the relative uncertainty ($\Delta H_{rel}$) to analyze the model's choice when faced with a conflict.

of relative uncertainty, derived from the confidence gap between modalities

By analyzing the model's outputs across benchmarks, we uncovered a clear and predictable pattern. As we systematically increased the preference uncertainty of one modality relative to the other, the model's probability of following that modality showed a consistent **monotonic decrease**. This finding confirms that modality following is not a fixed attribute but a fluid behavior that predictably shifts with the relative difficulty of unimodal inputs. However, we observed that a model does not necessarily follow the modality with the lower relative uncertainty. Instead, each model possesses a unique threshold—a subjective **balance point** of uncertainty that it is willing to tolerate. This balance point reveals the model's **inherent preference**. For example, a model with a strong inherent preference for vision might only follow the text if the text is *significantly* easier to process than the image.

We then sought to understand the internal mechanism *why* does a model hesitate or average its choices when the relative uncertainty is near its subjective balance point? To explore this, we categorized conflict scenarios into two types: a **clear region**, where one modality is significantly dominant, and an **ambiguous region**, where uncertainties are balanced. By probing layer-wise predictions, we reveal that the model's hesitation is visible internally as **"oscillations"**, where the top prediction repeatedly switches between conflicting modalities. Crucially, we discover that this cognitive struggle is not uniform but strictly localized to the middle-to-late layers. While early layers process features relatively stably, the model repeatedly vacillates in these deeper layers

when facing ambiguity, directly explaining the externally observed indecision.

Finally, we apply our framework to the practical challenge of **preference steering** via Supervised Fine-Tuning (SFT). Our experiments reveal a critical "failure of easy-to-hard generalization": models trained solely on data where the target modality is already dominant ("easy" data) fail to learn robust modality following. We demonstrate that effective steering requires targeting the **boundary cases** identified by our relative uncertainty metric, providing a principled guideline for data selection in alignment tasks. In summary, this paper makes four key contributions:

- We decompose modality following behavior into two core components: case-specific **preference uncertainty** and the model's stable **inherent preference**.

- Leveraging both **novel controllable** and **real-world datasets**, we establish a consistent empirical regularity: modality following probability monotonically decreases as preference uncertainty increases, allowing us to quantify inherent preference as the **balance point**.

- We uncover an internal "oscillation" mechanism localized to **deeper layers**, revealing that conflict resolution is a high-level process linking internal dynamics to external hesitation.

- We apply our framework to preference steering via fine-tuning, demonstrating that data efficiency is governed by preference uncertainty, thereby offering concrete guidance for future data selection.

## 2. Related Work

**Macro-level Modality Bias and Benchmarks.** Prior research has extensively characterized modality bias in MLLMs using aggregate statistics, such as the Text-Following Ratio (TFR) (Deng et al., 2025; Zhang et al., 2025). While benchmarks like $MC^2$ and MMIR reveal that preferences vary across models and tasks (Yan et al., 2025), they often treat datasets as monolithic, overlooking the **instance-specific reasoning confidence** that drives individual decisions.

**Attribution and Mechanistic Interpretability.** Efforts to explain modality preferences often focus on external factors like input order (Deng et al., 2025) or internal inconsistencies in learned representations (Zhu et al., 2024; Golovanevsky et al., 2025). Others utilize attribution methods to quantify modality influence (Parcalabescu & Frank, 2024). However, these approaches primarily offer a static view of "what" the model relies on, rather than the **dynamic computational process** of "how" it resolves ambiguity. We fill this gap by investigating the layer-wise internal dynamics that characterize model hesitation during inference.

**Modality Reliance and Pre-training Priors.** The tendency toward modality imbalance is often linked to the "visual priors" LLMs acquire during text-only pre-training (Han et al., 2025). While previous studies identify these as global, static biases (Gat et al., 2021; Frank et al., 2021), they do not explain how such **intrinsic preferences** are modulated by real-time difficulty during runtime. Our study disentangles these pre-existing biases from fluid reasoning confidence, providing a unified view of MLLM decision-making. Detailed related work is provided in Appendix B.

## 3. Defining Conflicting Inputs and Quantifying Modality Following

**Conflicting Inputs.** We define a *conflicting input* as a triplet $(I, T, Q)$ consisting of an image $I$, a textual description $T$, and a question $Q$, such that the unimodal predictions of the MLLM $M_\theta$ disagree:

$$Y_v = M_\theta(Q, I) \neq Y_t = M_\theta(Q, T).$$

Here, $Y_v$ and $Y_t$ denote the predictions when the model relies solely on the visual or textual modality, respectively. For example in Figure 1 (a), consider the question $Q$ = "What is the color of the square?". If the image $I$ shows a red square, while the text $T$ states "The color of the square is the same as a morpho butterfly's wings", then the image supports the answer "red" whereas the text suggests "blue". This forms a concrete instance of a conflicting input triplet $(I, T, Q)$. This setting requires the model to resolve contradictory cues and implicitly decide which modality to follow.

**Macro-level Metrics for Modality Following.** Given a conflicting input $x = (I, T, Q)$, the multimodal prediction is $Y_m = M_\theta(x)$. We categorize the outcome as **vision-following** if $Y_m = Y_v$, **text-following** if $Y_m = Y_t$, and **other** otherwise. To quantify the aggregate modality-following behavior on a dataset, we adopt the traditional approach of calculating following ratios. We define the text-following ratio (TFR) and vision-following ratio (VFR) as:

$$\text{TFR} = \frac{|\{x : Y_m = Y_t\}|}{|\{x : Y_m \in \{Y_v, Y_t\}\}|}, \quad \text{VFR} = 1 - \text{TFR}.$$

These ratios offer a simple, macro-level statistic of a model's aggregate behavior. In subsequent sections, we will deconstruct how these statistics emerge from a deeper interplay between case-specific uncertainty and a model's inherent preference, which these ratios alone cannot capture.

## 4. Experimental Setup: Diverse Benchmarks and Uncertainty Quantification

To provide a rigorous foundation for investigating how **modality following** is governed by preference uncertainty and inherent preference, we establish a comprehensive experimental framework. This section details: (1) our multi-tier dataset strategy that combines **fine-grained controllable synthesis** with **complex real-world benchmarks**, and (2) the validation of **entropy** as a robust, modality-comparable uncertainty metric.

### 4.1. Constructing Diverse and Controllable Conflict Scenarios

To ensure the universality of our subsequent analysis, we evaluate MLLMs across a diverse suite of datasets spanning the spectrum from systematically isolated scenarios to complex real-world benchmarks.

**Controllable Synthetic Datasets.** Existing benchmarks often struggle to independently vary the reasoning difficulty of each modality. To overcome this, we first built a **Synthetic Information Conflict Dataset** (including Color and Attribute Recognition) where each instance is defined by a task type $\mathcal{T}$ and two integer-based *design tiers*, $d_v$ and $d_t$. These tiers allow us to systematically manipulate complexity: **Vision tier** ($d_v$)**:** Modulates perceptual difficulty by adding distractors, shrinking targets, or introducing occlusions (e.g., from a single clear square to a small, obscured object in a crowded scene).**Text tier** ($d_t$)**:** Controls reasoning depth, ranging from direct conflicting statements to multi-hop relational descriptions (e.g., "the same color as a morpho butterfly's wings"). By pairing these tiers, we generate a structured landscape of conflict cases with predictable relative difficulty.

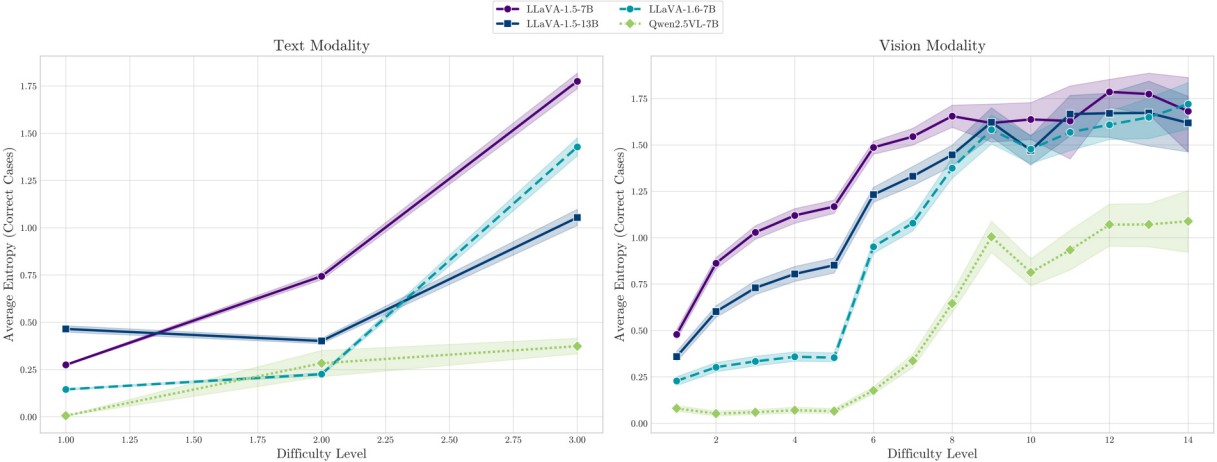

*Figure 2.* **Unimodal Entropy Trends Across Difficulty Tiers in Controllable Synthetic Datasets**. Average unimodal entropy for text (left) and vision (right).

**Real-world and Adversarial Benchmarks.** To complement these controlled environments, we expand the experimental scope to include two modality conflict benchmarks representing **authentic, complex scenarios**: the $MC^2$ **Benchmark** (Zhang et al., 2025) and the **Modified** $Pascal\ VOC$ **dataset** (Hua et al., 2025). By incorporating these established datasets alongside our controllable synthetic tasks , we establish an evaluation landscape that spans the full spectrum from systematically isolated mechanisms to representative real-world challenges. This duality ensures that our framework is validated across both controllable benchmarks and authentic multimodal contexts. Detailed specifications of the datasets are provided in Appendix C.

### 4.2. Quantifying Perceived Uncertainty via Entropy

To capture the model's **intrinsic preference uncertainty** rather than relying on human-defined difficulty, we primarily employ the Entropy of its output distribution over the answer token. For any unimodal input $x$, the uncertainty is $H(x) = -\sum_{y\in\mathcal{V}} p(y \mid x) \log p(y \mid x)$. While entropy serves as our primary unified metric across modalities ($H^{(v)}, H^{(t)}$), we also verify that the fundamental behaviors remain robust across alternative metrics like negative log-probability and prediction margin(see Appendix F.4).

**Validation of Uncertainty Coverage.** Leveraging the high degree of controllability in our synthetic dataset, we analyze unimodal entropy trends (Figure 2) to ensure the data provides a representative and fine-grained coverage of model-perceived uncertainty: Entropy increases monotonically with higher design tiers ($d_v, d_t$), confirming that our controlled difficulty tiers accurately induce varying levels of model-centric uncertainty. Both modalities span a broad and comparable dynamic range (from near-zero to over 1.75), demonstrating that the dataset successfully cov-

ers the full spectrum of reasoning difficulty rather than being limited to trivial cases. The differences in entropy levels across models (e.g., lower entropy for Qwen2.5-VL) align with their known performance scaling, providing a robust, model-grounded foundation for the subsequent analysis of multimodal conflict resolution.

## 5. Modality Following is Shaped by Relative Uncertainty

**Contradictory Behaviors at the Macro Level.** To thoroughly investigate modality following, we evaluated a diverse suite of MLLMs across three distinct datasets: our proposed controllable dataset, the $MC^2$ benchmark, and the modified $Pascal\ VOC$ dataset. Our model selection covers the LLaVA-1.5 (Liu et al., 2024a) and LLaVA-1.6 families (Li et al., 2024), the Qwen-VL series (Wang et al., 2024; Qwen et al., 2025; Bai et al., 2025), as well as GLM-4V (GLM et al., 2024) and CogVLM2 (Hong et al., 2024). Detailed specifications of these models are provided in Appendix D.

Initial macro-level analysis using the **Text-Following Ratio (TFR)** reveals a striking divergence among models: while the CogVLM2 and LLaVA series appear predominantly text-following, the GLM-4V and Qwen-VL series are notably vision-following (Figure 3a). Such starkly different aggregate behaviors on the same datasets raise a fundamental question: *Does this reflect a fixed, arbitrary preference, or is it governed by a deeper, yet-unseen principle?*

**A Finer Lens: Relative Unimodal Uncertainty.** Static macro-level ratios overlook case-specific reasoning confidence. To capture this, we introduce **relative unimodal uncertainty** ($\Delta H_{\text{rel}}$), the normalized difference be-

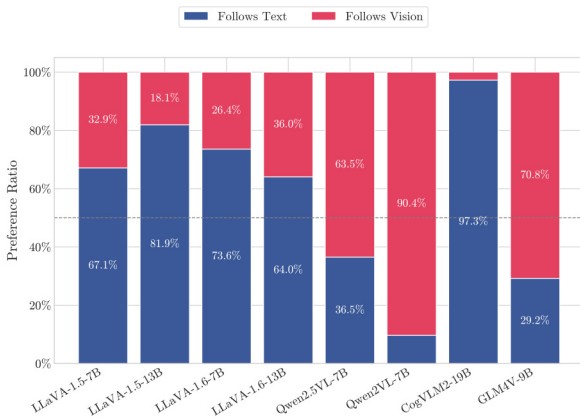

*(a)* Overall macro-level performance

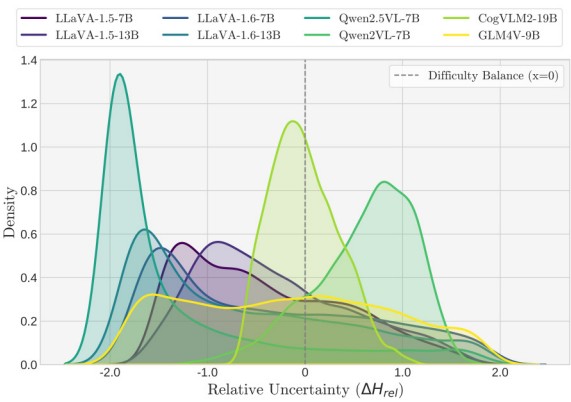

*(b)* Relative uncertainty distribution

*Figure 3.* Macro-level modality-following ratios and relative uncertainty distributions of model performance on the dataset.

tween unimodal entropies derived from decoupled components (image $I$ or text $T$ with question $Q$): $\Delta H_{\text{rel}}(x) = \frac{2\left(H^{(t)}(x) - H^{(v)}(x)\right)}{H^{(t)}(x) + H^{(v)}(x)}$ where $H^{(t)}$ and $H^{(v)}$ represent text-only and vision-only uncertainty, respectively. A negative $\Delta H_{\text{rel}}$ indicates higher confidence in the text modality, while a positive value favors vision. Crucially, as shown in Figure 3b, most models exhibit a **highly similar distribution** of $\Delta H_{\text{rel}}$, skewed significantly toward text-confidence, yet some of their final modality following choices remain diametrically opposed. This **asymmetry between input uncertainty and macro-level output** deepens the mystery: if different models perceive the same "text-is-easier" trend in the data, why do their aggregate preferences diverge so starkly? This paradox suggests that macro-level statistics are a confounded reflection of two distinct factors: the data's relative difficulty and the model's own **inherent preference**. To disentangle these, we must look beyond aggregate distributions and uncover the dynamic law governing their interplay.

**A Consistent Monotonic Trend.** The answer emerges when we shift our perspective from aggregate statistics to the dynamic relationship between uncertainty and choice. By plotting the probability of a model following the text modality against the corresponding $\Delta H_{\text{rel}}$ for each case, the apparent chaos resolves into a single, unified pattern, as shown in Figure 4a. For all eight models, regardless of architecture or scale, the curve shows a smooth, **monotonic decrease**. In other words, as text becomes harder relative to vision (i.e., as $\Delta H_{\text{rel}}$ increases), the probability that the model follows the text steadily and predictably decreases. This discovery directly confirms our central hypothesis from the Introduction: modality following is not a fixed trait but a dynamic behavior governed by relative preference uncertainty. The trend is further robust across alternative uncertainty indices, such as negative log-probability and prediction margin, confirming its metric-agnostic nature. Detailed results are provided in Appendix F.4.

**Quantitative Verification.** To statistically corroborate the visual monotonicity, we compute Spearman's rank correlation coefficient ($\rho$) between $\Delta H_{\text{rel}}$ and the text-following probability for each of the 72 model-task pairs (8 models $\times$ 9 tasks across our benchmarks). $\rho$ is uniformly negative across all 72 pairs, with 65/72 ($\sim$90%) falling below $-0.8$ and many reaching $-0.99$ or lower (e.g., $\rho = -1.00$ for LLaVA-1.5-7B on MC$^2$ activity recognition and for LLaVA-1.6-7B / CogVLM2-19B on Pascal VOC). The remaining seven pairs correspond to tasks with a compressed $\Delta H_{\text{rel}}$ range or models saturated near one extreme; the most extreme is GLM-4V-9B on Pascal VOC color recognition ($\rho = -0.37$), which still remains negative. The full per-model, per-task breakdown is provided in Appendix G.

**Quantifying Inherent Preference via the Balance Point.** While all models obey this consistent trend, their curves are positioned differently along the axis. This leads to our second key insight. We define the **balance point** as the $\Delta H_{\text{rel}}$ value at which the model is equally likely to follow either modality (a 50% text-following probability). This balance point provides a principled, quantitative measure of the model's **inherent modality preference**. A balance point below zero indicates an inherent *vision preference* (as text must be significantly easier to be treated as equal), while a point above zero indicates an inherent *text preference*. This allows us to disentangle a model's fluid, in-the-moment decision-making from its stable, underlying biases.

**Reconciling Macro-Level Contradictions.** Our framework, which separates unimodal capability (reflected in the $\Delta H_{\text{rel}}$ distribution) from inherent preference (the balance point), can now fully explain the apparent contradictions from our initial macro-level analysis. Consider Qwen2-VL, which appears more vision-following than Qwen2.5-VL

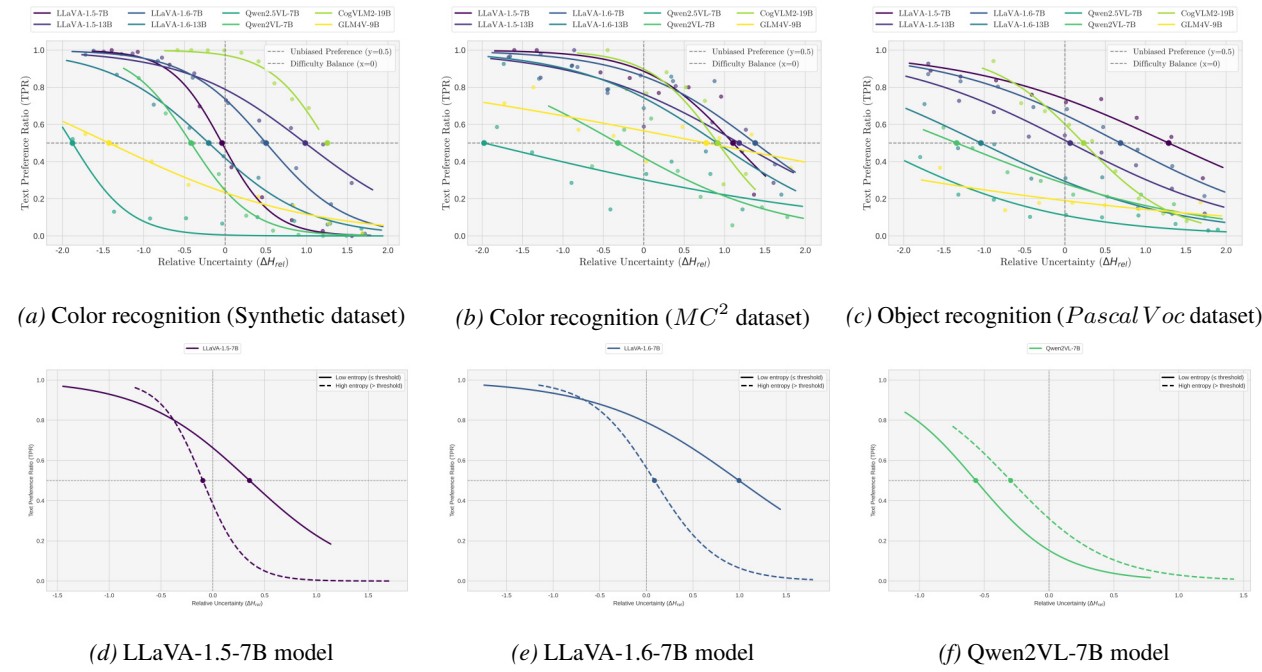

*(a)* Color recognition (Synthetic dataset)    *(b)* Color recognition ($MC^2$ dataset)    *(c)* Object recognition ($Pascal\,Voc$ dataset)

*(d)* LLaVA-1.5-7B model    *(e)* LLaVA-1.6-7B model    *(f)* Qwen2VL-7B model

*Figure 4.* **Universality and Robustness of the Relative Uncertainty Law across Datasets and Entropy Levels.** (a-c) General Monotonicity: The monotonic decrease of TPR is consistent across diverse datasets ; (d-f) Robustness to Absolute Uncertainty: The monotonic decrease of TPR is consistent across different absolute entropy levels in color recognition task (Synthetic dataset).

based on its VFR. Our analysis reveals this is largely a dataset artifact. Qwen2-VL's stronger visual capabilities on this specific dataset mean that more data points simply fall into the "vision-is-easier" (positive $\Delta H_{\text{rel}}$) region, mechanically inflating its vision-following stats. However, Qwen2.5-VL has a balance point further to the left (more negative), revealing a *stronger inherent vision preference*, as it continues to trust vision even when text is substantially easier. Similarly, the difference between LLaVA and Qwen models is not just about capability. While both face a dataset where text is often easier, Qwen models possess a clear inherent vision preference (negative balance point), whereas LLaVA models have a neutral or text-leaning preference (balance point near or above zero). It is this crucial difference in their *inherent preference* that drives their divergent behaviors, a nuance entirely missed by macro-level metrics.

**Generalization across Diverse Scenarios.** As illustrated in Figure 4(a-c), the consistent monotonic relationship between relative uncertainty ($\Delta H_{rel}$) and modality following remains robust across the Synthetic dataset, $MC^2$ and Modified Pascal VOC benchmarks. Notably, despite the stark disparities in image distributions and textual description styles across these domains, the model's behavior is governed by the same underlying trend. This cross-dataset consistency demonstrates that the observed preference dynamics reflect a general property of MLLM conflict resolution rather than artifacts of specific data distributions. Furthermore, this monotonic pattern extends consistently to six additional $MC^2$ task categories (activity, attribute, object, sport recognition, positional reasoning, and sentiment understanding) as well as to attribute-binding and prompt-diversified settings, all consolidated in Appendix F.2.

**Robustness to absolute difficulty** To rule out artifacts of specific difficulty levels, we split the data into high-entropy (hard) and low-entropy (easy) subsets based on median total entropy. As illustrated in Figure 4(d-f), both subsets independently preserve the strict monotonic decline. Notably, high-entropy cases (dashed lines) exhibit steeper curves with balance points closer to zero compared to low-entropy cases (solid lines). This aligns with the intuition that uncertain models are more easily swayed by relative difficulty, whereas confident models show greater resistance. Additional comparisons are presented in Appendix F.3.

## 6. The Internal Mechanism: Oscillation in the Face of Ambiguity

**Probing Layer-wise Predictions** To uncover the mechanisms behind the established behavioral regularity, we probe the model's internal reasoning dynamics. Our analysis reveals that external hesitation near the balance point is a direct consequence of internal **oscillations** between conflicting modalities. We categorize scenarios into two distinct regions based on relative uncertainty: an *ambiguous re-*

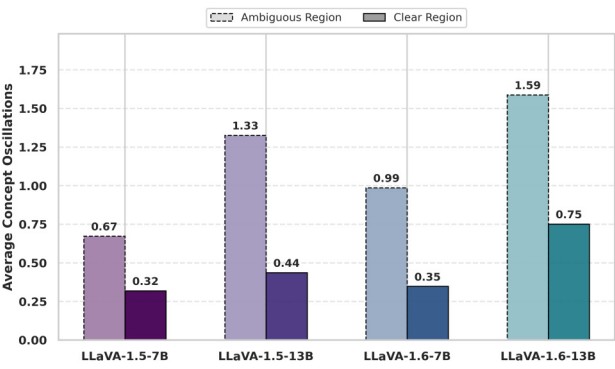

*(a)* Average Oscillations in ambiguous and clear region.

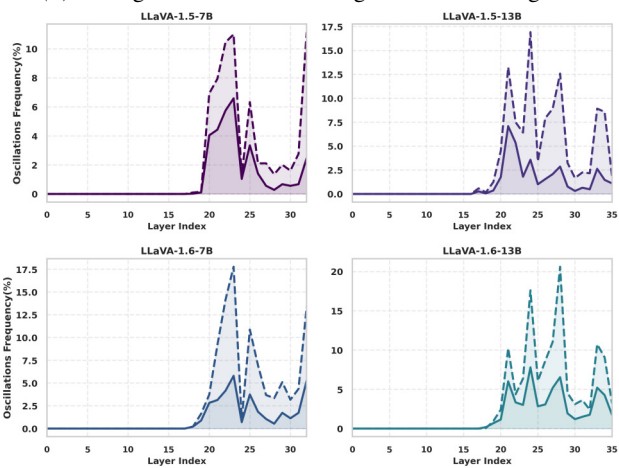

*(b)* Oscillations Frequency across model layers

*Figure 5.* Visualization of internal concept oscillations across different models. **(a) Average Oscillation Counts:** The average number of concept oscillations per sample. Lighter bars with dashed borders represent the **Ambiguous Region**, while darker solid bars represent the **Clear Region**. **(b) Layer-wise Frequency Distribution:** Illustrate the oscillation frequency (%) across the model's layers. **Dashed lines** correspond to the Ambiguous Region, and **solid lines** correspond to the Clear Region.

gion, defined as cases where $|\Delta H_{\mathrm{rel}} - \text{balance point}| \leq 0.5$ (i.e., modalities are similarly uncertain), and a *clear region*, where $|\Delta H_{\mathrm{rel}} - \text{balance point}| > 0.5$ (i.e., one modality is substantially more confident). We tracked the top-1 prediction at each layer during forward using a *LogitLens*-style technique (nostalgebraist, 2020; Zhang et al., 2024) and defined **oscillations** as switches between vision-supported and text-supported answers.

The comparative results are presented in Figure 5a. The bar charts reveal a stark contrast: the oscillation frequency in the ambiguous region is nearly double that in the clear region across models. Figure 5b further illustrates the distribution of these oscillations across layers. We observe that for all models, oscillations predominantly initiate in the **middle-to-late stages** (typically after layer 15). Based on these observations, we derive two key conclusions: (1) Models exhibit

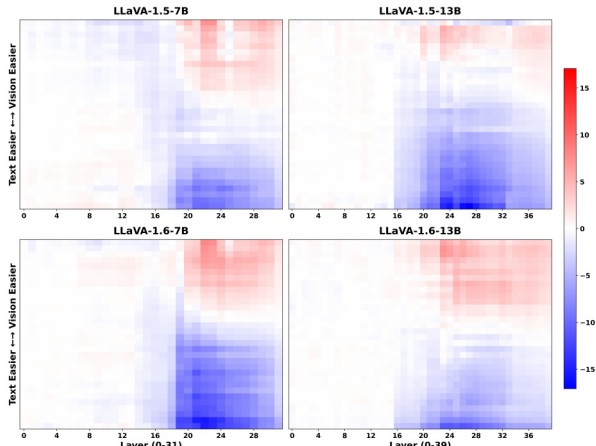

*Figure 6.* Logit Difference Heatmap Across Model Layers and Relative Uncertainty.

significantly higher instability when relative uncertainty is high in ambiguous regions. (2) This "hesitation" is not a global phenomenon but is strictly localized to the deeper layers of the network, suggesting that conflict resolution is a high-level cognitive process. To validate the generality of these findings, we extended this analysis to all evaluated models across three additional datasets. As detailed in Appendix F.5, these extensive experiments demonstrate striking consistency in behavioral patterns, confirming that this layer-wise oscillation mechanism is a robust property of MLLMs when resolving multimodal ambiguity.

**Visualizing Indecision with Logit Difference Heatmaps.** Figure 6 plots the logit difference (text-supported minus vision-supported logits) across layers for four representative models. Here, the x-axis denotes the layer index, and the y-axis represents relative uncertainty ($\Delta H_{\mathrm{rel}}$). Crucially, the "white" neutral zones (indicating near-zero logit differences) do not strictly align with the geometric center ($y = 0$) but rather correspond to each model's unique **balance point** identified in Figure 4a. For instance, while LLaVA-1.5-7B and LLaVA-1.6-13B exhibit indecision near the center, the white bands for LLaVA-1.5-13B and LLaVA-1.6-7B shift noticeably towards the vision-easier region (positive $\Delta H_{\mathrm{rel}}$). This confirms that internal hesitation is driven by the model's subjective equilibrium. Conversely, regions distant from this balance point exhibit early saturation to deep red or blue, signaling that once outside the ambiguous zone, models commit quickly and stably. A more detailed analysis of these internal trajectories can be found in Appendix I.

The layer-wise oscillation counts and the logit-difference heatmaps show that internal hesitation is tightly aligned with each model's balance point rather than the geometric center.

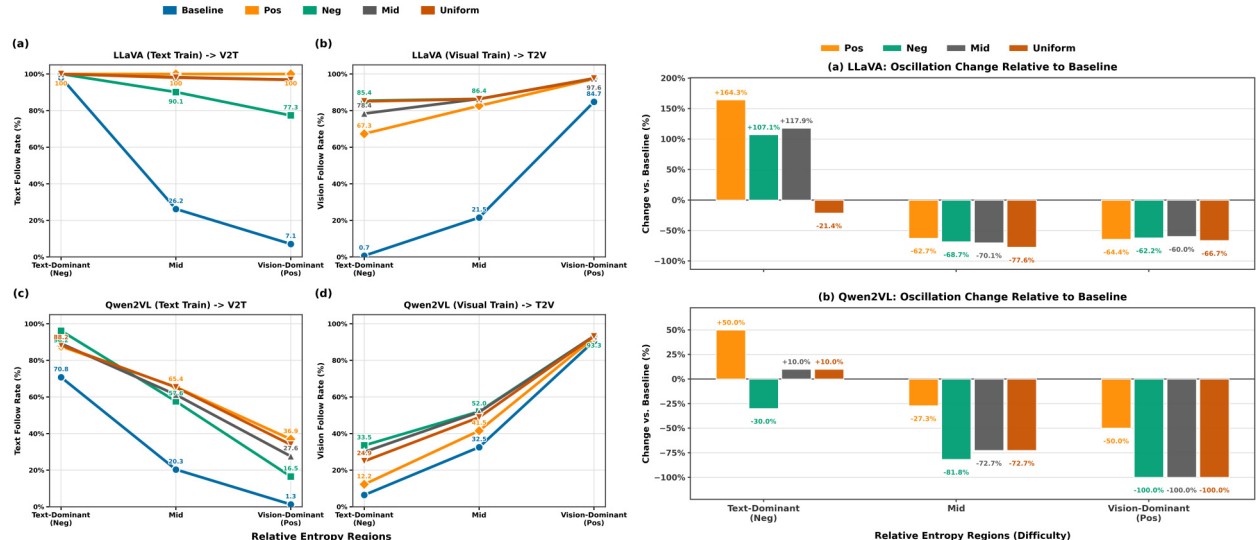

*(a)* Target Modality Follow Rate of Finetuned Models.      *(b)* Oscillation Change Ratio vs Baseline.

*Figure 7.* **Impact of Preference Finetuning and Internal Dynamics.** **(a)** The modality following probability curves under different data selection strategies (Pos, Neg, Mid, Uniform) across LLaVA15-7B and Qwen2VL-7B. **(b)** The relative change in internal "oscillation" (conflict) compared to the baseline. Negative values indicate a reduction in hesitation.

**Causal Validation.** To confirm that the observed oscillations are specifically elicited by modality *conflict* rather than general model instability, we measured oscillation frequency in a matched-modality condition, where image and text convey *consistent* information. The oscillation count in this agreement condition is only 19% of that in the conflict condition, demonstrating that internal hesitation arises specifically from contradictory multimodal inputs. Then to directly test whether middle-to-late layer oscillations *drive* the final modality choice, we perform an activation-patching experiment: the activations at oscillating middle-to-late layers are replaced with those from a non-conflicting "clear" condition, thereby suppressing the oscillatory dynamics. This intervention results in a **70% reduction** in oscillation frequency and raises modality-following rates to **95%**, closely matching behavior on unambiguous inputs. These causal results confirm that the layer-wise oscillations are not merely correlated with indecision but are a direct mechanism underlying the model's final modality choice.

## 7. Application: Guiding Data Selection for Preference Steering

### 7.1. Validation on Controllable Dataset

**Motivation and Setup.** To explore how relative uncertainty guides preference steering, we conducted a controlled Supervised Fine-Tuning (SFT) experiment. We first split the dataset into training and testing sets, ensuring both contained samples across the full entropy spectrum. The training set was then used to construct four distinct subsets: three

based on specific difficulty regions—**Pos** (Vision-Easier), **Neg** (Text-Easier), and **Mid** (Ambiguous)—and one **Uniform** set sampled globally. Models (LLaVA-1.5-7B and Qwen2-VL-7B) were fine-tuned to switch their modality preference (e.g., Vision → Text) and evaluated on the common stratified test set. Detailed dataset construction and training hyperparameters are provided in Appendix E.

**Results.** Figure 7a reports the target modality following rates on the test set after fine-tuning with training sets of varying relative uncertainty. Our observations are three-fold: (1) **General Efficacy:** Compared to the baseline (blue line), all fine-tuned models show an improved propensity to follow the target modality across all regions. (2) **Consistency with Baseline Difficulty:** The post-tuning performance distribution mirrors the baseline's inherent difficulty. Even after finetuning, models consistently exhibit the lowest following rates in regions where the target modality was originally weakest ("hard" regions). For example, when steering towards Text (shown in panel a/c in Figure 7a), performance remains lowest in the Pos region (originally Vision-dominant). (3) **Inefficiency of "Easy" Data:** Training exclusively on data where the target modality is already dominant ("easy" data) proves to be the least efficient strategy for achieving general improvement. Our results reveal that models trained on these "easy" data struggle to generalize to complex scenarios. In the Vision-steering task (shown in panel b/d in Figure 7a), the **Pos-trained model** (orange line) learned from easy, vision-dominant samples, consistently exhibits the poorest performance in the challenging Neg region.

**Mechanism.** Figure 7b offers a mechanistic explanation for the inefficiency of "Easy" data by tracking post-tuning oscillation frequency changes (using Visual-steering as the primary example). First, we observe a distinct contrast between stabilization and struggle. In regions where the target modality is naturally easier or balanced (Pos and Mid), fine-tuning consistently yields a substantial reduction in oscillations (negative bars), signaling a confident commitment to the new preference. Conversely, in the **Hard (Neg) region**, oscillation frequencies show minimal reduction or even noticeable increases. This reflects a state of cognitive dissonance, where the model vacillates between its inherent prior and the newly tuned instruction. Crucially, this instability is exacerbated by "Easy" training. The **Pos-trained model** which only saw easy visual dominance during training—exhibits the poorest oscillation control in the challenging Neg region. For instance, it triggers the most dramatic spike in oscillations (e.g., +164% for LLaVA), whereas models trained on harder data (e.g., Neg) manage to better suppress this instability. This confirms that training on trivial samples fails to equip the model with robust conflict-resolution mechanisms, leaving it prone to high-frequency vacillation when facing difficulty.

**Practical Implications.** Highlighting the failure of easy-to-hard generalization, our results suggest that data efficiency in preference alignment (e.g., SFT) hinges on preference uncertainty. Future strategies should therefore prioritize ambiguous and hard cases over trivial cases that merely reinforce existing priors.

### 7.2. Real-World Validation on DriveBench

A natural concern is whether the boundary-case principle holds up outside controlled benchmarks. To address this, we transfer the framework to **DriveBench** (Xie et al., 2025), a multi-camera autonomous-driving benchmark covering object-centric perception, prediction, and planning—a safety-critical setting in which misleading textual cues are especially harmful.

**Setup.** For each object-centric perception question, we construct answer-targeted conflict captions for every non-ground-truth option, then stratify training examples by relative entropy into three subsets that mirror the *Pos/Mid/Neg* convention of our main SFT experiment: **Text-easier** (misleading text is more confident than vision), **Tie** (text and vision exhibit comparable uncertainty), and **Vision-easier** (vision is more confident than the misleading text). To probe generalization beyond multiple choice, we additionally evaluate an open-ended reasoning split, in which sequence-level entropy is the mean per-token entropy, conflict captions are generated by Gemini-3.1-Pro, and Gemini-3-Flash serves as the automatic judge.

**Results.** First, the untrained **Base** model collapses under conflict: LLaVA-1.5-7B drops from 76.67% vision-only accuracy to just 3.33% under misleading captions, and Qwen2.5-VL-7B from 83.33% to 6.67% (full MCQ table in Appendix H, Table 8), confirming that real driving systems are highly vulnerable to text-vision conflicts. Second, training on the **Text-easier** subset where the misleading text appears most convincing, consistently yields the largest recovery. On MCQ perception, Text-easier training lifts LLaVA-1.5-7B to **50.00%** and Qwen2.5-VL-7B to **63.33%**, clearly above the Tie and Vision-easier subsets. The same ordering holds in the more demanding long-form reasoning setting (Table 1), where Text-easier training produces the highest overall scores for both models—most strikingly for Qwen2.5-VL-7B (32.81 → **45.35**).

*Table 1.* **Long-form Reasoning on DriveBench (Conflict Input Setting).** Overall scores are produced by Gemini-3-Flash acting as an automatic judge.

| Model | Training subset | Overall Score (↑) |
|---|---|---|
| LLaVA-1.5-7B | Base | 13.93 |
| | Tie | 20.10 |
| | Vision-easier | 18.25 |
| | Text-easier | **22.88** |
| Qwen2.5-VL-7B | Base | 32.81 |
| | Tie | 37.32 |
| | Vision-easier | 34.64 |
| | Text-easier | **45.35** |

The ranking *Text-easier > Tie > Vision-easier* is exactly the ordering predicted by our synthetic experiments: training on samples whose preference uncertainty places them near the model's decision boundary transfers most effectively to held-out conflicts. This is real-world evidence that the boundary-case principle is not an artifact of toy benchmarks but extends to safety-critical multimodal applications. The full MCQ table and additional details on benchmark scope, caption construction, stratification thresholds, and the long-form evaluation protocol are provided in Appendix H.

## 8. Conclusion

This work establishes that modality following in MLLMs is a dynamic process governed by **relative preference uncertainty** and **inherent preference**. We observe a consistent monotonic trend: the probability of following a modality decreases predictably as its relative uncertainty grows, with the introduced **balance point** serving as a capability-agnostic metric for preference. Then, we identify middle-to-late layer **oscillations** as the internal driver of model indecision in ambiguous regions. By disentangling intrinsic bias from unimodal capabilities and highlighting the criticality of **boundary cases** for alignment, providing practical guidance for developing robust MLLMs.

## Acknowledgements

This work is partially supported by the MBZUAI Research Fund BF0100. This work is also supported in part by the funding BAS/1/1689-01-01, RGC/3/7125-01-01, FCC/1/5940-20-05, FCC/1/5940-06-02, and King Abdullah University of Science and Technology (KAUST) – Center of Excellence for Generative AI, under award number 5940 and a gift from Google.

## Impact Statement

This paper presents work whose goal is to advance the field of Machine Learning. There are many potential societal consequences of our work, none of which we feel must be specifically highlighted here.

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

## A. The Use of Large Language Models (LLMs)

During the preparation of this paper, we used large language models (LLMs) solely as general-purpose writing assistants. Specifically, LLMs were employed to help refine the clarity, grammar, and readability of our drafts, as well as to suggest alternative phrasings in English. Importantly, all conceptual contributions including the design of research questions, development of methods, execution of experiments, and interpretation of results were conceived and carried out entirely by the authors. The authors carefully reviewed and edited all text suggested by LLMs to ensure accuracy and originality, and we take full responsibility for the final content of the paper.

## B. Extended Related Work

**Characterizing Conflicting Information.** A significant body of research has focused on how MLLMs behave when faced with contradictory inputs. Various benchmarks have been developed to probe this phenomenon, revealing that many models exhibit a "blind faith" in text, systematically ignoring visual information (Deng et al., 2025). However, this tendency is not universal, as other studies demonstrate that preferences vary significantly across different models and scenarios (Zhang et al., 2025; Liu et al., 2024b). Further work with benchmarks like MMIR has focused on the model's ability to detect and reason about such inconsistencies (Yan et al., 2025). Our work unifies these observations by proposing a framework that moves beyond dataset-level statistics to case-specific relative preference uncertainty.

**Explaining and Interpreting Conflict Resolution.** One line of research seeks to explain modality preference via external factors, such as the order of inputs (Deng et al., 2025) or instructional prompts. Others delve deeper, attributing behavior to internal factors like inconsistencies within the model's learned knowledge representations (Zhu et al., 2024; Golovanevsky et al., 2025). A third approach uses attribution methods, such as those based on Shapley values, to quantify the relative influence of each modality (Alishahi et al., 2019; Parcalabescu & Frank, 2024). Despite these insights, the dynamic, layer-by-layer computational process remains under-explored. Our work introduces the concept of internal "oscillations" as observable evidence of the conflict resolution process, explaining the mechanics of model hesitation.

**Modality Reliance and Pre-training Priors.** Research has revealed a frequent tendency toward modality imbalance. (Gat et al., 2021) introduced the **Perceptual Score** to quantify reliance, revealing that models often exploit textual shortcuts. Similarly, (Frank et al., 2021) used cross-modal input ablation to diagnose information flow, uncovering significant asymmetries between visual and textual recruitment. Complementing these behavioral diagnoses, (Han et al., 2023) analyze *which* visual inputs are more memorable to machines, showing that the saliency of visual evidence to a model is itself an asymmetric property that shapes downstream multimodal behavior. The origins of textual dominance are further elucidated by (Han et al., 2025), who show that LLMs acquire "visual priors" solely from text pre-training. More recently, (Endo & Yeung-Levy, 2026) dissect perception versus reasoning bottlenecks in small multimodal models, indicating that errors under multimodal inputs cannot be attributed to a single capability axis but instead reflect an interaction between perceptual fidelity and reasoning competence. Our work advances this line by treating these pre-training priors and capability asymmetries as an inherent preference baseline (the balance point), which is dynamically modulated by the case-specific confidence gap between modalities during inference.

## C. Dataset Construction

### C.1. Real-world Information Conflict Datasets

To validate the generalization of our findings across broader and more realistic tasks, we adapted two real-world information conflict datasets—MC$^2$ (Zhang et al., 2025) and the synthetic dataset used by (Hua et al., 2025) into formats compatible with our methodology.

MC$^2$ is a benchmark dataset designed to evaluate the modality preferences of multimodal large language models (MLLMs). It consists of 2,000 samples sourced from TDIUC(Kafle & Kanan, 2017), covering eight types of perceptual tasks such as color detection, counting, and object recognition. The dataset was constructed through a semi-automated process:

- Images, questions, and corresponding visual answers were first collected from TDIUC;

- Large language models were employed to generate conflicting textual contexts and distracting answers that contradict the visual content;

- Baseline MLLMs were used to filter samples that could be correctly understood via a single modality;

- Multiple rounds of human validation were conducted to ensure genuine inter-modal conflicts and distinct answers supported by each modality.

Each case includes an image, a conflict text context, a question, and two modality-specific answers (text-based and vision-based). For our study, we selected cases from the color detection and object recognition tasks and further filtered them to include only those with single-word answers, so as to avoid potential ambiguities in answer evaluation.

The other real-world information conflict dataset we used was adapted from (Hua et al., 2025), which synthesized a dataset based on Pascal VOC (Everingham et al., 2010) using an adversarial sampling approach. The core methodology involves:

- Selecting a real image from the original dataset;

- Randomly sampling an incorrect class label that differs from the ground-truth label;

- Using a standardized template to generate a caption that contradicts the image content;

- Constructing multiple-choice options to systematically create evaluation samples with inconsistent image-text information.

Following this sampling and synthesis procedure, we constructed 2,470 multiple-choice samples from the Pascal VOC train subset. Unlike the original study, which employed templates with explicit image or text-biased instructions, we used several neutral question templates to ensure fair treatment of both textual and visual information. Table 2 presents the templates used in our dataset construction.

*Table 2.* Templates for Modified Pascal Dataset

| **Caption templates** |
| --- |
| This is an image of a {CLASS_LABEL}. |
| This is a photo of a {CLASS_LABEL}. |
| An image of a {CLASS_LABEL}. |
| A photo of a {CLASS_LABEL}. |
| This is a {CLASS_LABEL}. |
| A {CLASS_LABEL}. |
| **Question templates** |
| What is being shown? |
| What is presented? |
| What is depicted? |
| What is this? |
| **Option template** |
| Select from the following classes: |
| **Instruction template** |
| Answer the question using a single word or phrase. |
| **Answer template** |
| Image_Answer: |
| Text_Answer: |

## C.2. Synthetic-scenario information conflict dataset

To investigate the external performance and internal mechanisms of multimodal models when dealing with conflicts between image and text information, we constructed two Synthetic-scenario information conflict datasets. The first is **Color**

**Recognition Dataset**, which requires the model to identify the color of geometric shapes placed on a white canvas. The second is **Attribute Recognition Dataset**, adapted and filtered from the CLEVR(Johnson et al., 2017) dataset, whose task is to identify the material and shape of three-dimensional objects. Both datasets contain multiple task groups. Each group provides images with increasing visual complexity and text descriptions that contradict the image information while exhibiting increasing textual reasoning complexity. By systematically controlling the visual perception complexity ($d_v$) and the textual reasoning complexity ($d_t$), this design constructs conflict scenarios with diverse visual-textual difficulty combinations in a systematic manner.

### C.2.1. DATASET OVERVIEW

The Color Recognition Dataset consists of 400 groups, each containing 14 images and questions with 3 different types of conflict descriptions. Images with difficulty levels 0–4 are 800×600 pixels, while those with levels 5–13 are 224×224 pixels. The text is divided into three different types, with an average length of 22.7 words. In each group, the same image_answer color can be derived from any image information, while the same text_answer color which is different from the image_answer, can be obtained from any conflict description in the text. The distribution of image_answer and text_answer is as follows:

- **Image_answer Colors:** Red(67), Yellow(67), Blue(67), Green(66), Purple(66), Orange(67)

- **Text_answer Colors:** Red(67), Yellow(66), Blue(67), Green(67), Purple(66), Orange(67)

The Shape subset and the Material subset of the Attribute Recognition Dataset each contain 300 groups. Each group includes 4 images and questions with 3 different types of conflict descriptions. All images are 480×320 pixels, while the text is divided into five different types, with an average length of 30.0 words. In each group, the same image_answer attribute can be derived from any image information, while the same text_answer attribute which is different from the image_answer can be obtained from any conflict description in the text. The distribution of image_answer and text_answer is as follows:

- **Image_answer Shapes:** Sphere(108), Cube(100), Cylinder(92)

- **Text_answer Shapes:** Sphere(100), Cube(92), Cylinder(108)

- **Image_answer Materials:** Metal(160), Rubber(140)

- **Text_answer Materials:** Metal(140), Rubber(160)

### C.2.2. IMAGE GENERATION OF COLOR RECOGNITION DATASET

For each set of 14 images with a progressive difficulty gradient in the Color Recognition Dataset, we used the Python PIL library for rendering. The following is the generation pipeline.

1. **Initialization:** A **target shape** (e.g., Circle) is randomly selected.

2. **Color Assignment:**
   - **Visual Answer Color:** One color is randomly assigned to the target shape.
   - **Textual Answer Color:** A different color is randomly selected as the conflicting textual statement.

3. **Distractor Generation:** Distractor shapes are randomly chosen from the set excluding the target shape. Their colors are randomly selected from the set excluding both the visual and textual answer colors.

4. **Difficulty Tiers** ($d_v = 0$ **to** 13)**:** Fourteen progressive difficulty levels are defined by target size, number of distractors and occlusion. Parameters are specified in Table3.

*Table 3.* Visual Difficulty ($d_v$) Tiers Specification

| Difficulty($d_v$) | Target Size | # Distractors | Occlusion Rule |
|---|---|---|---|
| 0 | 80-200 pixels | 0 | No occlusion |
| 1 | 80-200 pixels | 1 | No occlusion |
| 2 | 80-200 pixels | 2 | No occlusion |
| 3 | 80-200 pixels | 3 | No occlusion |
| 4 | 80-200 pixels | 4 | No occlusion |
| 5 | 20%-40% of image | 7 | 50% occlusion rate |
| 6 | 20%-40% of image | 10 | 80% occlusion rate |
| 7 | 5%-10% of image | 7 | 50% occlusion rate |
| 8 | 5%-10% of image | 11 | 80% occlusion rate |
| 9 | 4%-6% of image | 20 | 30% occlusion rate |
| 10 | 4%-6% of image | 30 | 60% occlusion rate |
| 11 | 4%-6% of image | 40 | 50% occlusion rate |
| 12 | 4%-6% of image | 55 | 60% occlusion rate |
| 13 | 4%-6% of image | 70 | 70% occlusion rate |

**Note 1:** "Occlusion rate" refers to the proportion of distractors that visually overlap the target. Different rates for odd/even tiers introduce finer-grained difficulty variation.

### C.2.3. IMAGE SELECTION OF ATTRIBUTE RECOGNITION DATASET

All images in the Attribute Recognition Dataset were curated from the CLEVR dataset, which contains objects defined by three geometric shapes (cube, sphere, cylinder), two materials (rubber, metal), and eight colors. For each target attribute corresponding to the subset, our selection procedure began by forming all possible attribute–color pairs via the Cartesian product. For each unique pair, we identified images from the CLEVR validation set containing *exactly one* object matching that specific combination. The selected images were then assigned a difficulty level based on scene complexity, with a fixed number of images sampled per level to construct the final task groups. Table 4 shows the various difficulty levels of the pictures.

*Table 4.* Difficulty levels for image selection

| Difficulty($d_v$) | Number of objects in scene | Target object size |
|---|---|---|
| 0 | 3–4 objects | large |
| 1 | 6–8 objects | large |
| 2 | 6–8 objects | small |
| 3 | ≥10 objects | small |

### C.2.4. TEXTUAL MODALITY CONSTRUCTION

The conflict text issues between the Color Recognition Dataset and the Attribute Recognition Dataset share many similarities in terms of structure and pipeline construction. In both cases, we gradually increase the complexity of the textual modality by increasing the number of reasoning steps and converting explicit reasoning into implicit reasoning. The questions within the same group share a fixed **target_shape** with the images of that group, inquire an **attribute** depending on the dataset they belong to, and utilize an identical **text_answer** that contradicts the image information. Each textual problem follows the format of: [Conflict Description] + [Question] + [Command].

- **Question:** `What {attribute} is the {target_shape}?`

- **Command:** `Please use one word to answer this question.`

For each group, we generate 3 types of conflict description for Color Recognition Dataset and 4 for Attribute Recognition Dataset with increasing difficulty. The Table 5 below lists each type and a concise description, where **A** denotes the target_object, **T** denotes the text_answer, **B/S1/S2** represent randomly selected objects absent from the image, **D** represents a

real-world instance unambiguously possessing attribute T, and **Pos1/Pos2** denote a pair of opposite spatial relations Left and Right.

*Table 5.* Question types and descriptions (descriptions only)

| Difficulty($d_t$) | Type | Description |
|---|---|---|
| x | Original | No interference description. |
| 0 | Direct | The A is T. |
| 1 | Indirect_simple | The A's {attribute} is the same as a B. The B is T. |
| 2 | Indirect | The A's {attribute} is the same as a D. |
| 3 | Space(Attribute Recognition Dataset only) | There is a T S1, on the Pos1 of the S1 is a S2. The A's {attribute} is the same as the object Pos2 to the S2. |

**Robustness Processing:** To prevent models from solving tasks via superficial pattern matching, texts in Color Recognition Dataset for $d_t \geq 0$ were paraphrased using Qwen-Plus(Alibaba Cloud / QwenLM, 2025). This process preserved core semantics, reasoning structure, and key information tokens while varying sentence structure, prepositional phrases, and lexical choices.

**Control Group Setup:**For ablation studies, two types of control data were constructed:

- **Text-Irrelevant:** The target shape 'A' in conflict description only is replaced with a randomly chosen **non-target shape** (e.g., if target is 'circle', replace with 'triangle' or 'rectangle').

- **Image-Irrelevant:** The target shape 'A' in the entire text is replaced with a shape **never present** in the images ('star', 'cone', 'frustum'), maintaining the correspondence between the question and the text description while severing the connection with the image.

---

**Rewrite Questions Task**

====SYSTEM====
You are a conservative paraphrasing assistant specialized in subtle wording changes. Your goal is to rewrite a single question sentence while preserving *all* facts, *all* explicit instructions, and the exact multi-hop reasoning structure (number of inference steps and intermediate referents). Make only minor wording, grammar, punctuation, and token-count adjustments; do NOT add, remove, or transform factual content or the logical chain.

====USER====

Field type:
{FIELD_TYPE}
Original question:
{ORIGINAL_QUESTION}
Rewrite Instructions (STRICT):
1. Output exactly one rewritten question sentence (no explanation, no notes, no extra punctuation before/after).
2. Preserve *all* factual propositions and named referents. Do not add or remove facts.
3. Preserve the multi-hop reasoning structure:
- If the original is a single-step (direct), keep it single-step.
- If it is implicit multi-step (indirect), keep it implicit and do not make steps explicit.
- If it is explicit multi-hop (indirect_simple), keep the same explicit chain of premises and the same number of hops.
4. Preserve any explicit answering instruction exactly (e.g., "Please use one word to answer this question.").
5. Do not change the identity of entities (e.g., "hexagon", "pine tree", "circle") or the target attribute (e.g., "color").
6. Only rewrite wording, punctuation, and sentence flow to be more natural or shorter, and optionally reduce/increase token count slightly. You can use near-synonyms with very high similarity.
7. Avoid introducing pronouns that obscure referents; keep clarity of which object each premise references.
8. If the original contains multiple sentences that together form the multi-hop chain, you may combine or split them only if you exactly preserve the same premises and hop order.
Output: the single rewritten question sentence (no extra text).

### C.3. Illustrative Samples from the dataset

To provide a more intuitive understanding of our image-text conflict dataset, we have sampled several image-question pairs from the Color Recognition Dataset, the Attribute Recognition Dataset subsets, $MC^2$ and modified Pascal dataset, and presented them in Figure8, Figure9, Figure10, Figure11 and Figure12.

**MC2 195894**

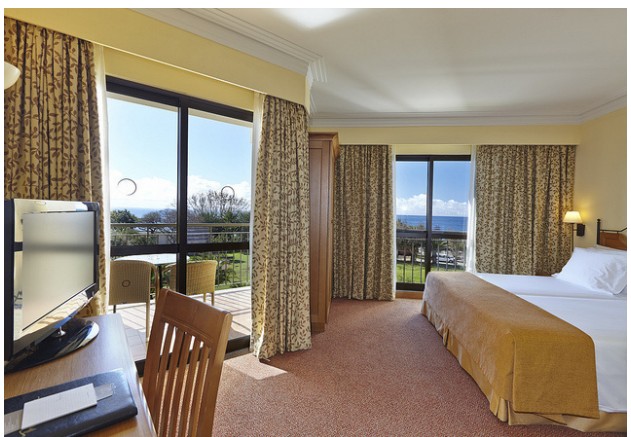

**Caption:** The bedroom ... The sheets on the large bed are made of a deep blue fabric ...

**Question:** What color is the sheet?
**Command:** Please use one English word to answer this question.
**Vision-based Answer:** *white*
**Text-based Answer:** *blue*

**MC2 518363**

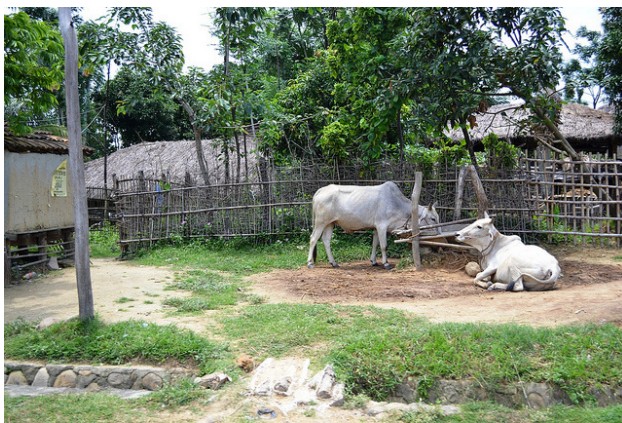

**Caption:** Two eagles perched on a fence in a backyard, with ...

**Question:** What animal is shown in the photo?
**Command:** Please use one English word to answer this question.
**Vision-based Answer:** *cow*
**Text-based Answer:** *eagle*

*Figure 8.* A selection of image-text pairings from MC$^2$ Dataset. The text highlighted in red indicates the descriptions and answers that conflict with the image information.

**Pascal Group730**

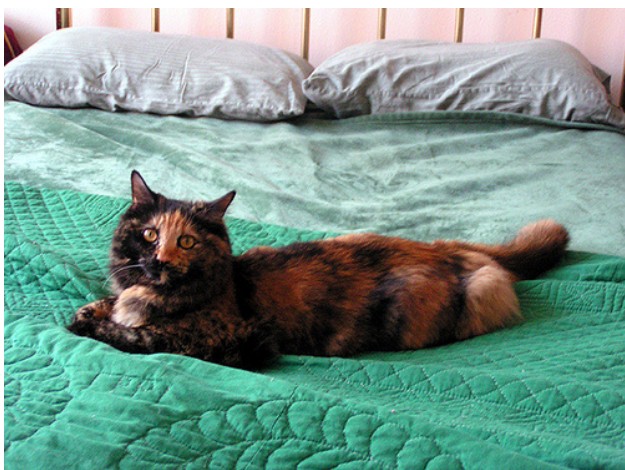

**Caption:** An image of a dog.
**Selections:** Select from the following classes: cow, potted plant, dog, bicycle, cat.
**Question:** What is this?
**Command:** Answer the question using a single word or phrase.
**Vision-based Answer:** *cat*
**Text-based Answer:** *dog*

**Pascal Group1453**

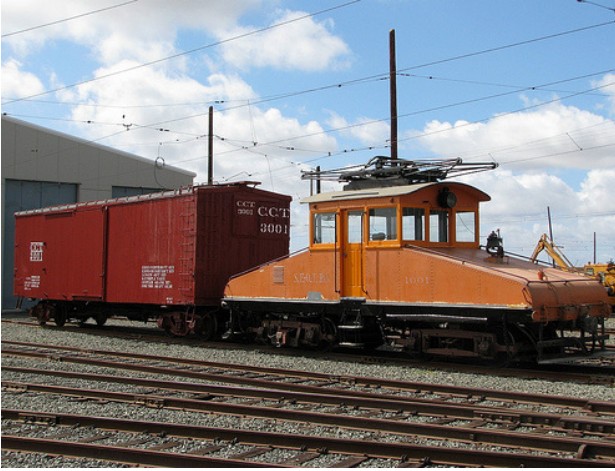

**Caption:** A photo of a bird.
**Selections:** Select from the following classes: sheep, train, bird, sofa, potted plant.
**Question:** What is being shown?
**Command:** Answer the question using a single word or phrase.
**Vision-based Answer:** *train*
**Text-based Answer:** *bird*

*Figure 9.* A selection of image-text pairings from modified Pascal VOC Dataset. The text highlighted in red indicates the descriptions and answers that conflict with the image information.

**Group41 Difficulty0**        **Group41 Difficulty3**

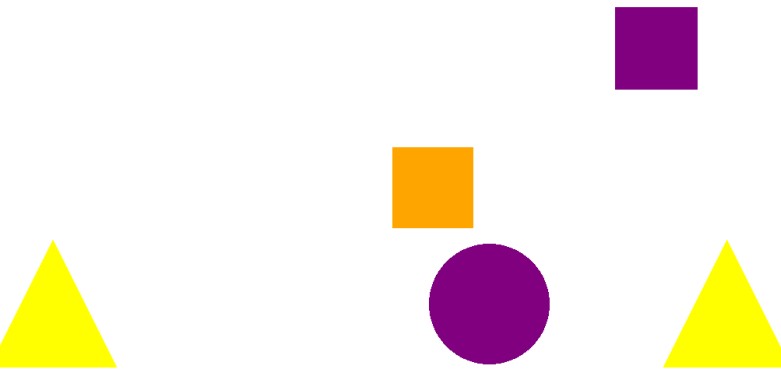

**Original:**
**Question:** What color is the triangle?
**Command:** Please use one word to answer this question.
**Vision-based Answer:** *Yellow*
**Text-based Answer:**

**Direct:** The triangle is blue.
**Question:** What color is the triangle?
**Command:** Please use one word to answer this question.
**Vision-based Answer:** *Yellow*
**Text-based Answer:** *Blue*

**Group41 Difficulty6**

**Group41 Difficulty15**

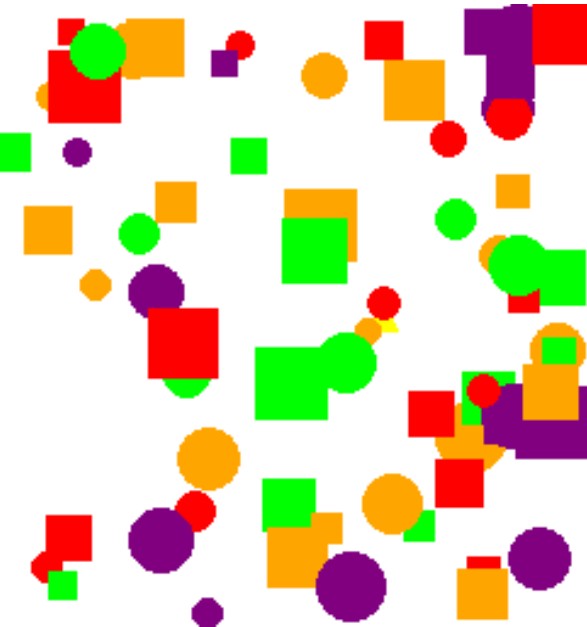

**Indirect_simple:** The triangle's color is the same as a pentagon. The pentagon is blue.
**Question:** What color is the triangle?
**Command:** Please use one word to answer this question.
**Vision-based Answer:** *Yellow*
**Text-based Answer:** *Blue*

**Indirect:** The triangle's color is the same as a mailbox in the US.
**Question:** What color is the triangle?
**Command:** Please use one word to answer this question.
**Vision-based Answer:** *Yellow*
**Text-based Answer:** *Blue*

*Figure 10.* A selection of image-text pairings from a group in the Color Recognition Dataset. The text highlighted in red indicates the descriptions and answers that conflict with the image information.

**Group193 Difficulty0**

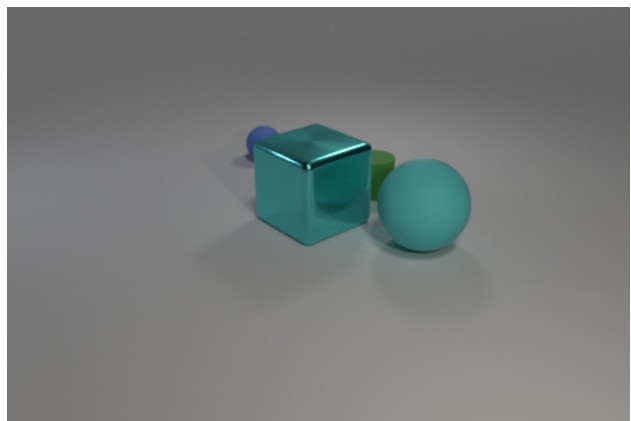

**Direct:** The cyan rubber object is a cylinder.

**Question:** What is the shape of the cyan rubber object?
**Command:** Please answer with one word.
**Vision-based Answer:** *sphere*
**Text-based Answer:** *cylinder*

**Group193 Difficulty2**

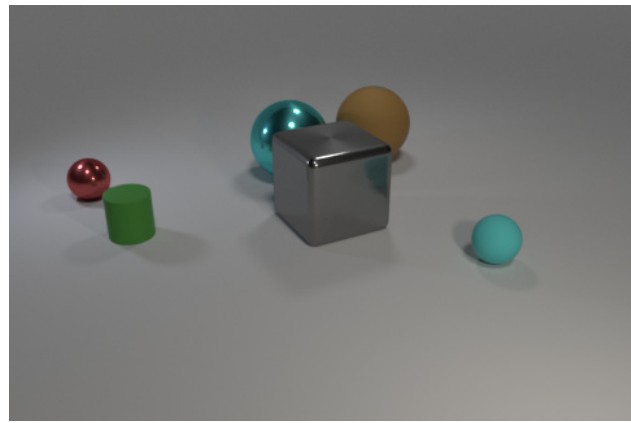

**Indirect:** The cyan rubber object's shape is the same as a log.
**Question:** What is the shape of the cyan rubber object?
**Command:** Please answer with one word.
**Vision-based Answer:** *sphere*
**Text-based Answer:** *cylinder*

*Figure 11.* A selection of image-text pairings from a group in the Shape subset of the Attribute Recognition Dataset. The text highlighted in red indicates the descriptions and answers that conflict with the image information.

**Group79 Difficulty1**

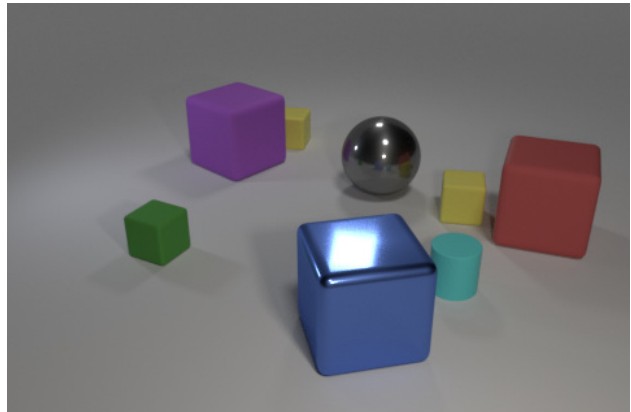

**Indirect_simple:** The Frustum is rubber, blue cube's material is the same as the Frustum.

**Question:** What is the material of the blue cube?
**Command:** Please use one word to answer this question.
**Vision-based Answer:** *metal*
**Text-based Answer:** *rubber*

**Group79 Difficulty3**

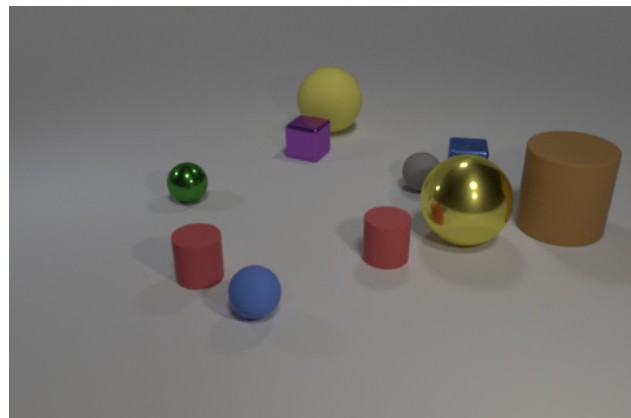

**Space:** There is a rubber cone, the right of the cone is a wood frustum. The blue cube's material is the same as the object left to the wood frustum.
**Question:** What is the material of the blue cube?
**Command:** Please use one word to answer this question.
**Vision-based Answer:** *metal*
**Text-based Answer:** *rubber*

*Figure 12.* A selection of image-text pairings from a group in the Material subset of the Attribute Recognition Dataset. The text highlighted in red indicates the descriptions and answers that conflict with the image information.

## D. Details of Evaluated Models

To ensure a comprehensive evaluation and verify the generalization capabilities of our findings, we selected four representative Vision-Language Model (VLM) families. These models were chosen to encapsulate diverse architectural strategies in visual encoding and modality alignment. Table 6 illustrates the key structural differences among these models.

*Table 6.* Overview of VLM families included in our experiments. *Deep Fusion* refers to modality interaction within LLM layers.

| Model | LLM Backbone | Visual Encoder | Resolution Strategy | Fusion Type |
|---|---|---|---|---|
| LLaVA-1.5 | Llama | CLIP-ViT-L | Fixed | MLP Projection |
| LLaVA-1.6 | Vicuna | CLIP-ViT-L | AnyRes | MLP Projection |
| Qwen2.5-VL | Qwen2.5 | SigLIP-based | Native Dynamic | MLP + M-RoPE |
| GLM-4v | GLM-4 | EVA-CLIP | High-Res Fixed | MLP Projection |
| CogVLM2 | Llama-3 | EVA-CLIP | High-Res Fixed | Deep Fusion |

## E. Appendix: Preference Steering Experiment Details

### E.1. Dataset Construction

To account for the unique uncertainty distribution of each model, we computed $\Delta H_{\text{rel}}$ using the respective base models (LLaVA-1.5-7B and Qwen2-VL-7B) individually. We utilized a consistent held-out testing set of **8,409 samples** for evaluation.

For the training data, we partitioned the samples into subsets based on a threshold of $\delta = 0.5$:

- **Pos (Vision-Easier):** $\Delta H_{\text{rel}} > 0.5$.

- **Neg (Text-Easier):** $\Delta H_{\text{rel}} < -0.5$.

- **Mid (Ambiguous):** $|\Delta H_{\text{rel}}| \leq 0.5$.

- **Uniform:** A random selection stratified to cover the full entropy spectrum.

To isolate the effect of data difficulty from data scale, we downsampled all training subsets to a unified size for each model: **1,614 samples** per subset for LLaVA-1.5 and **782 samples** per subset for Qwen2-VL.

### E.2. Training Setup

We performed Supervised Fine-Tuning (SFT) using LoRA with a rank of $r = 32$ and $\alpha = 32$. Models were trained for 1.5 epochs to ensure convergence without overfitting. The specific configurations were:

- **Qwen2-VL-7B:** Learning rate of $2e-5$. Target modules included attention layers: `["q_proj", "k_proj", "v_proj", "o_proj"]`.

- **LLaVA-1.5-7B:** Learning rate of $3e-5$. Target modules included both attention and MLP layers: `["q_proj", "k_proj", "v_proj", "o_proj", "gate_proj", "up_proj", "down_proj"]`.

## F. More experimental on various models and datasets

### F.1. Macro-level performance and Relative uncertainty distribution Measure

To evaluate the universality of our framework and findings, we extend our analysis to three additional task settings beyond those discussed in the main text: Color Recognition and Object Recognition on the $MC^2$ dataset, as well as Object Recognition on the Pascal VOC dataset. As detailed in Figure 13, we present:

- Macro-level Performance (Left Column): A summary of the aggregate modality-following behaviors (Text-Dominant vs. Vision-Dominant), illustrating the global preference tendencies of different models on these specific tasks.

- Relative Uncertainty Distribution (Right Column): The density plots of the relative preference uncertainty ($\Delta H_{\text{rel}}$).

Consistently, we observe that the distribution of uncertainty varies significantly across tasks and models, further necessitating the need for uncertainty-aware analysis.

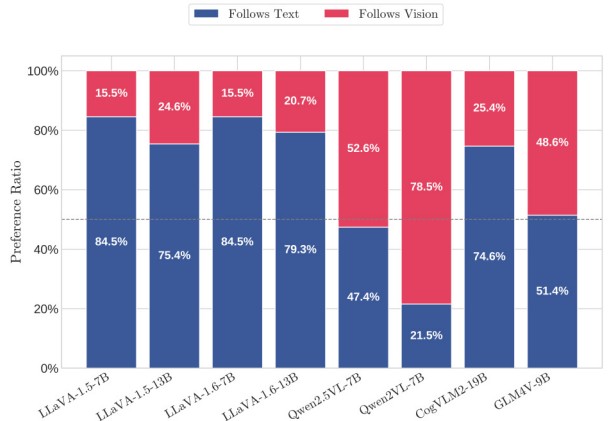

*(a)* Overall macro-level performance of color recognition task in $MC^2$ dataset

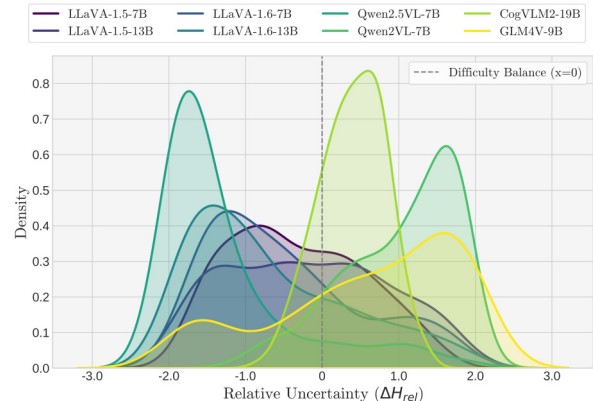

*(b)* Relative uncertainty distribution of color recognition task in $MC^2$ dataset

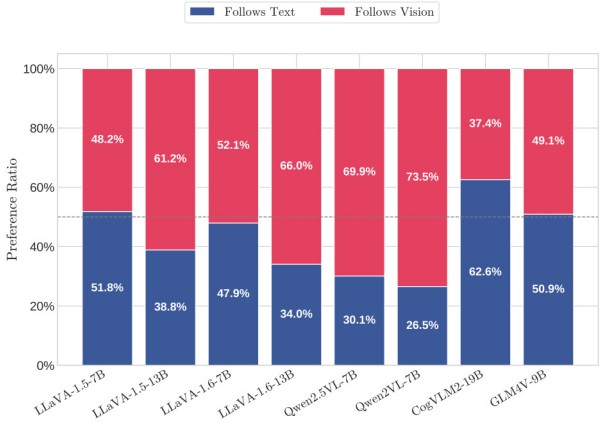

*(c)* Overall macro-level performance of object recognition task in $MC^2$ dataset

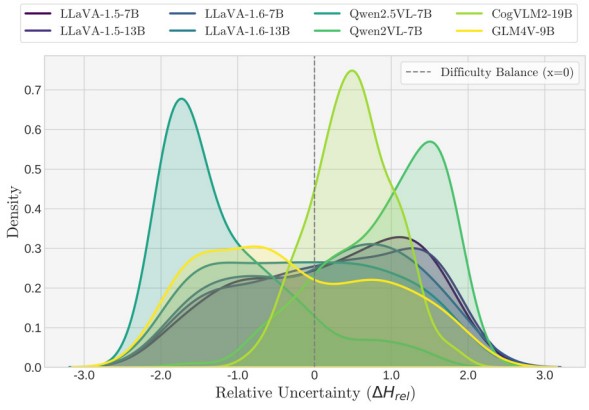

*(d)* Relative uncertainty distribution of object recognition task in $MC^2$ dataset

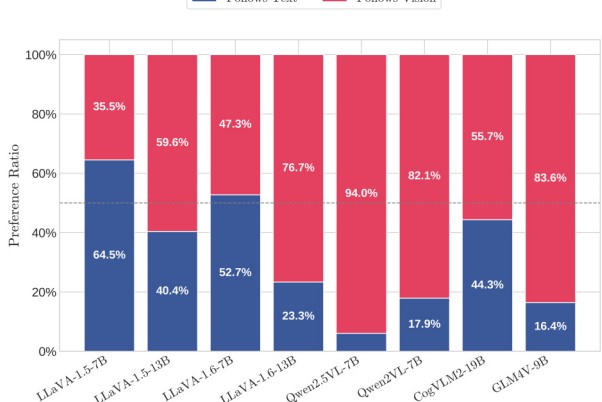

*(e)* Overall macro-level performance of object recognition task in $Pascal\,VOC$ dataset

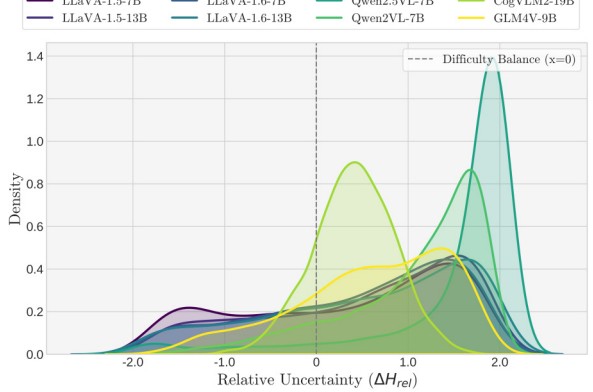

*(f)* Relative uncertainty distribution of object recognition task in $Pascal\,VOC$ dataset

*Figure 13.* Macro-level modality-following ratios and relative uncertainty distributions of model performance on the dataset.

## F.2. Consistency of Monotonicity: Generalization across Tasks and Settings

To further substantiate the consistency of the monotonic relationship between relative uncertainty and modality following, we extend our evaluation beyond the primary datasets discussed in the main text. Figure 14 presents eight additional settings grouped into two complementary axes of generalization:

- **Diverse $MC^2$ task categories (panels a–f):** Beyond the simple color recognition discussed in the main text, we evaluate six additional $MC^2$ tasks spanning perceptual recognition (activity, attribute, object, sport) and higher-level semantic reasoning (positional reasoning, sentiment understanding). The monotonic decrease of text-following probability with $\Delta H_{\mathrm{rel}}$ is preserved across all eight models on every task, demonstrating that the trend is not confined to low-level perception.

- **Generalization to Attribute Binding (panel g):** Using a subset from our Controlled Dataset focused on attribute binding (e.g., "the shiny metallic cube"), Figure 14g shows that the trend governs not just global perception but also fine-grained feature grounding.

- **Robustness to Prompt Phrasing (panel h):** To rule out the possibility that the model's preference is an artifact of specific sentence structures, we diversified the prompts of the Color Recognition task using Qwen-rewrites (Appendix C.2.4). Figure 14h demonstrates that the monotonic trend is invariant to linguistic variations.

Across all these varied dimensions—task type, reasoning complexity, and linguistic formulation—the consistent monotonic trend holds: models become less likely to follow a modality as their relative uncertainty regarding that modality increases.

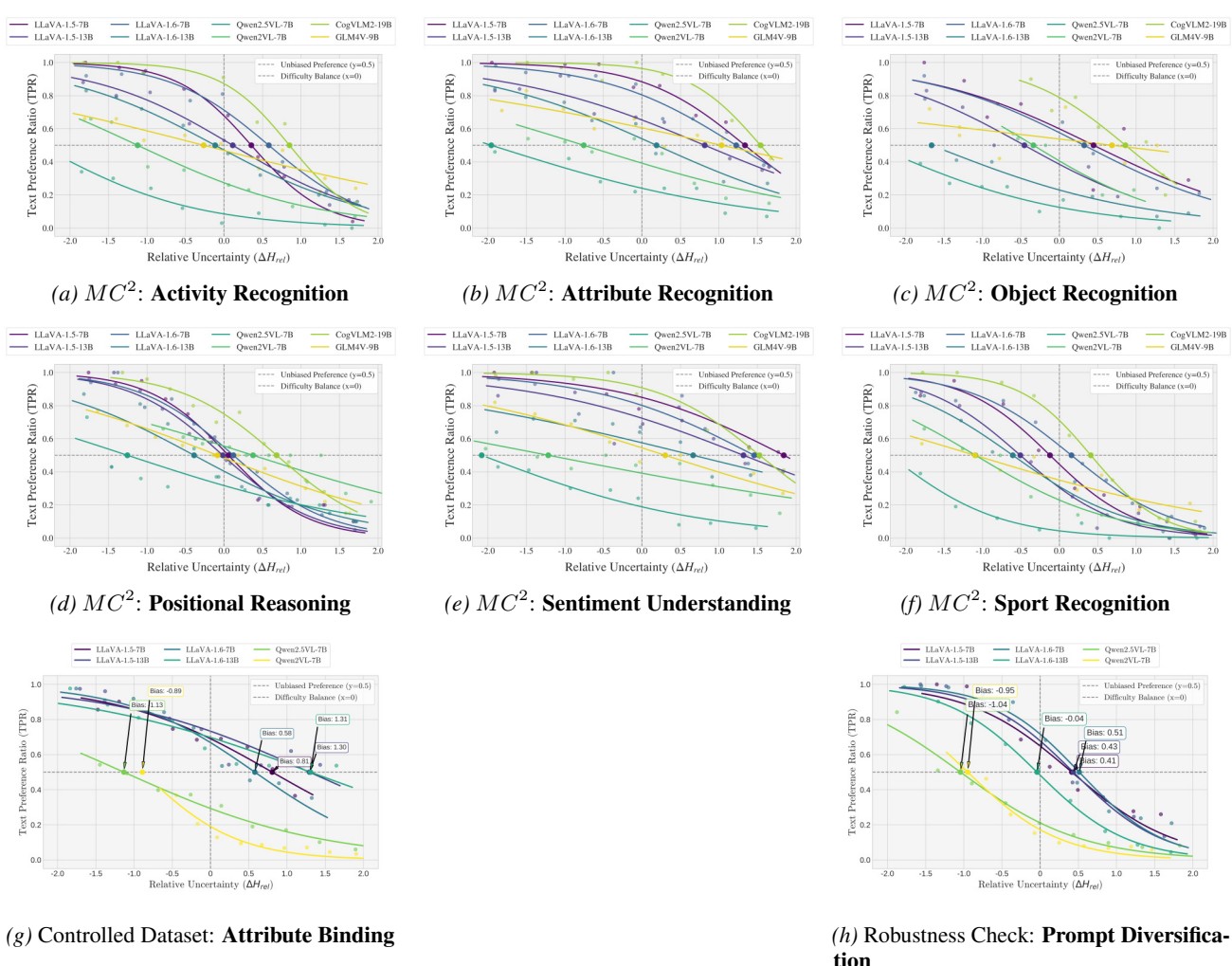

*(a) $MC^2$:* **Activity Recognition**    *(b) $MC^2$:* **Attribute Recognition**    *(c) $MC^2$:* **Object Recognition**

*(d) $MC^2$:* **Positional Reasoning**    *(e) $MC^2$:* **Sentiment Understanding**    *(f) $MC^2$:* **Sport Recognition**

*(g)* Controlled Dataset: **Attribute Binding**

*(h)* Robustness Check: **Prompt Diversifica-tion**

*Figure 14.* **Generalization of Modality Following Curves across Tasks and Settings.** Panels (a)–(f) show the text-following probability versus relative preference uncertainty ($\Delta H_{\text{rel}}$) across six $MC^2$ task categories, spanning perceptual recognition (activity, attribute, object, sport) and higher-level semantic reasoning (positional reasoning, sentiment understanding). Panels (g)–(h) further test generalization beyond $MC^2$: (g) attribute-binding tasks from our Controlled Dataset (e.g., "the shiny metallic cube") confirm the trend holds for fine-grained feature grounding; (h) prompt-diversified Color Recognition (Qwen-rewritten captions, Appendix C.2.4) confirms invariance to linguistic phrasing. Across all eight settings and all eight evaluated models, the same consistent monotonic decrease is observed; shifts in the balance points reflect each model's inherent preference under the specific data characteristics.

## F.3. Consistency of Monotonicity: High vs. Low Absolute Uncertainty

We extend our analysis of absolute difficulty to three additional large-scale models: **LLaVA-1.5-13B**, **LLaVA-1.6-13B**, and **Qwen2.5-VL-7B**. Using the same methodology as the main text, we partition the Controlled Dataset into high-entropy (dashed lines) and low-entropy (solid lines) subsets based on the median total entropy.

As shown in Figure 15, two key observations emerge:

- **Consistent Monotonicity:** Consistent with our empirical regularity, all three models independently preserve the strict monotonic decline of modality following probability as relative preference uncertainty increases, regardless of the absolute entropy level.

- **Impact of Absolute Difficulty:**
  - For **LLaVA-1.5-13B** and **Qwen2.5-VL**, the results align with our main findings: high-entropy (hard) cases exhibit steeper curves with balance points shifting closer to zero (the center). This confirms that high absolute uncertainty

typically makes models more susceptible to relative cues.

– **LLaVA-1.6-13B** presents a notable exception. While it adheres to the monotonicity law, its high-entropy curve does not significantly shift toward the center compared to the low-entropy curve. This suggests that LLaVA-1.6-13B possesses a more rigid internal calibration, maintaining stable preference dynamics even under high overall uncertainty.

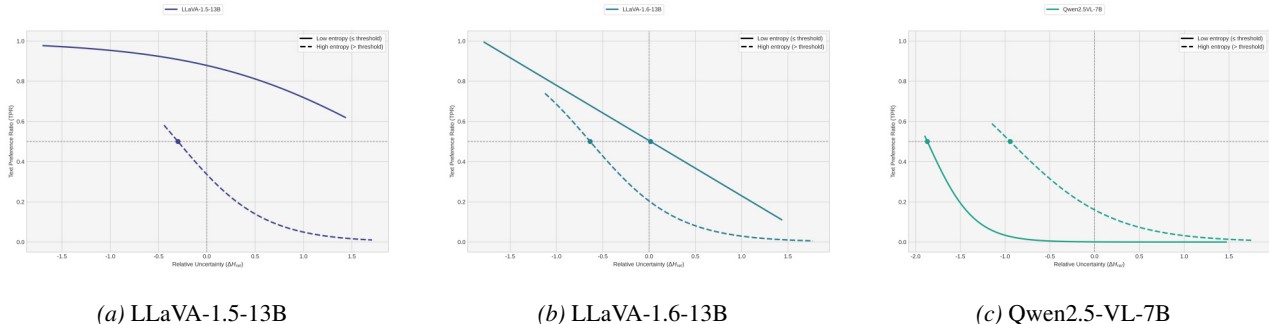

*(a)* LLaVA-1.5-13B        *(b)* LLaVA-1.6-13B        *(c)* Qwen2.5-VL-7B

*Figure 15.* **Robustness to Absolute Uncertainty across Additional Models.** Solid lines represent Low Entropy (Easy) subsets, and dashed lines represent High Entropy (Hard) subsets.

### F.4. Consistency of Monotonicity: the selection of different uncertainty indices

To verify whether Shannon Entropy is the unique prerequisite for our findings, or if the phenomenon reflects a deeper cognitive state independent of specific mathematical formulations, we extended our evaluation to two alternative uncertainty metrics:

1. **Prediction Margin (Gap):** Defined as $1 - (p_{\text{top1}} - p_{\text{top2}})$. This metric captures the ambiguity between the top two candidates, focusing on conflict intensity.

2. **Negative Log-Probability (NLL):** Defined as $-\log p(y|x)$. This metric directly reflects the model's surprisal or raw confidence in its chosen output.

**A Unified View.** Despite differences in calculation, Entropy, Margin, and NLL serve as unified proxies for the model's **reasoning hesitation**. High values in any of these metrics (or low margin) indicate that the model is struggling to distinguish between modalities.

**Results.** As illustrated in Figure 16, we plot the modality-following curves using these alternative metrics across our **Controlled Dataset**, $MC^2$ **Color Recognition**, and $Pascal$ **VOC Object Recognition**. Crucially, the **strict monotonic decline** is robustly preserved across all datasets and metrics. The curves exhibit the same characteristic shape: as relative uncertainty (regardless of how it is measured) increases, the model's adherence to a modality drops. This consistency confirms that the observed monotonic trend is driven by the underlying decision dynamics, avoiding the fragility often associated with single-metric analyses.

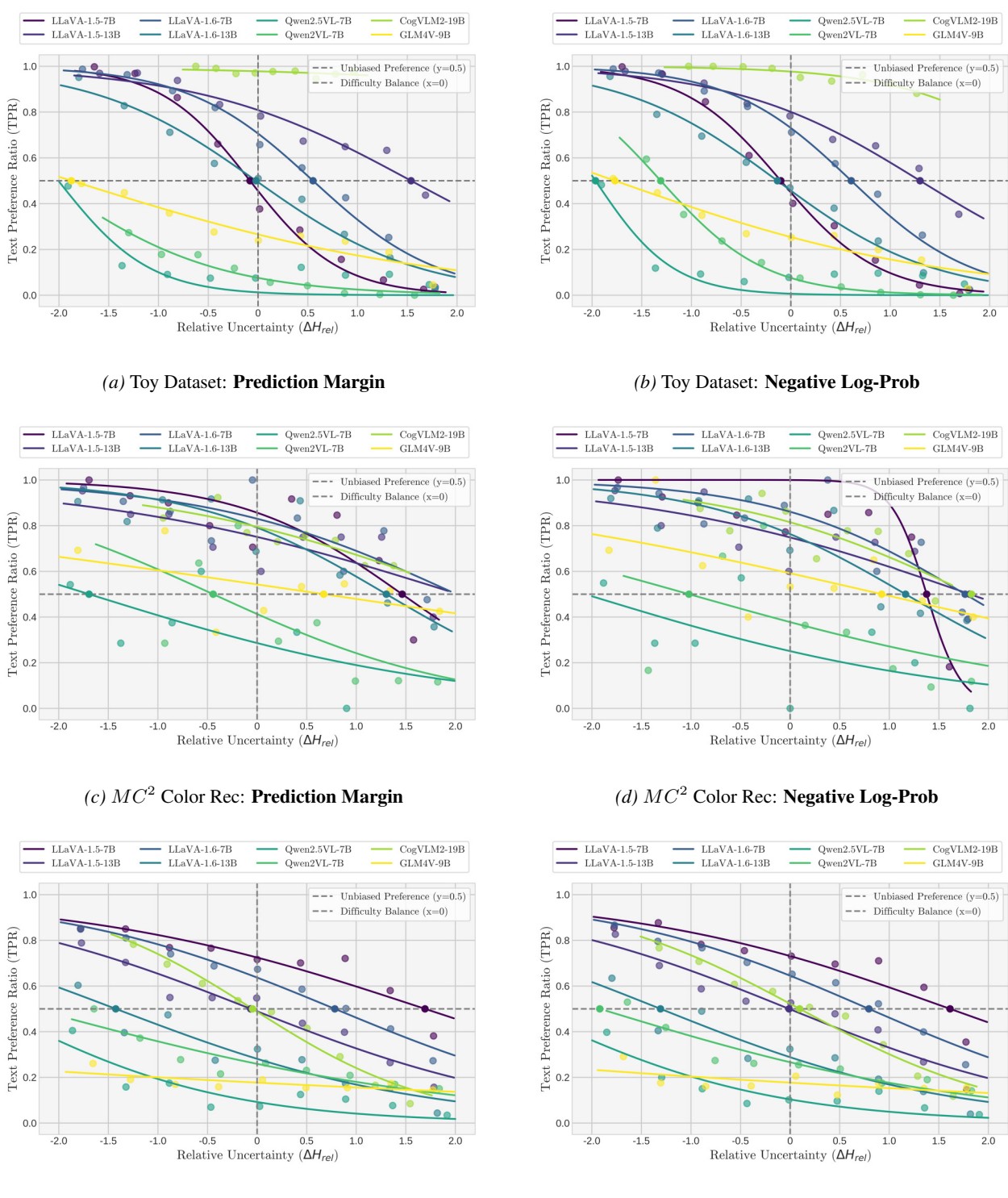

*(a)* Toy Dataset: **Prediction Margin**

*(b)* Toy Dataset: **Negative Log-Prob**

*(c)* $MC^2$ Color Rec: **Prediction Margin**

*(d)* $MC^2$ Color Rec: **Negative Log-Prob**

*(e) Pascal* VOC Obj Rec: **Prediction Margin**

*(f) Pascal* VOC Obj Rec: **Negative Log-Prob**

*Figure 16.* **Generalization across Uncertainty Metrics.** We replicate the analysis using Prediction Margin (Left Column) and Negative Log-Probability (Right Column) across different datasets. The preservation of the monotonic curve demonstrates that our findings are metric-agnostic and capture the fundamental uncertainty of the models.

### F.5. Universality of Layer-wise Oscillation Mechanism

To validate the generality of our findings regarding the internal conflict resolution mechanism, we extended the layer-wise analysis to all evaluated models across three additional datasets: $MC^2$ **Color Recognition**, $MC^2$ **Object Recognition**, and $Pascal$ **VOC Object Recognition**.

As visualized in Figure 17, Figure 18, and Figure 19, we consistently observe that:

- **High Oscillation in Ambiguity:** Across all tasks and models, the **Ambiguous** subsets (grey bars/lines) exhibit significantly higher oscillation magnitudes compared to the clear Regions.

- **Localized Conflict Resolution:** The layer-wise dynamics figure(right columns) confirm that this "hesitation" is primarily localized to the middle and late layers, where the model integrates conflicting modal information before reaching a final decision.

These extensive experiments confirm that the "oscillation" mechanism is a universal property of MLLMs when resolving multimodal ambiguity.

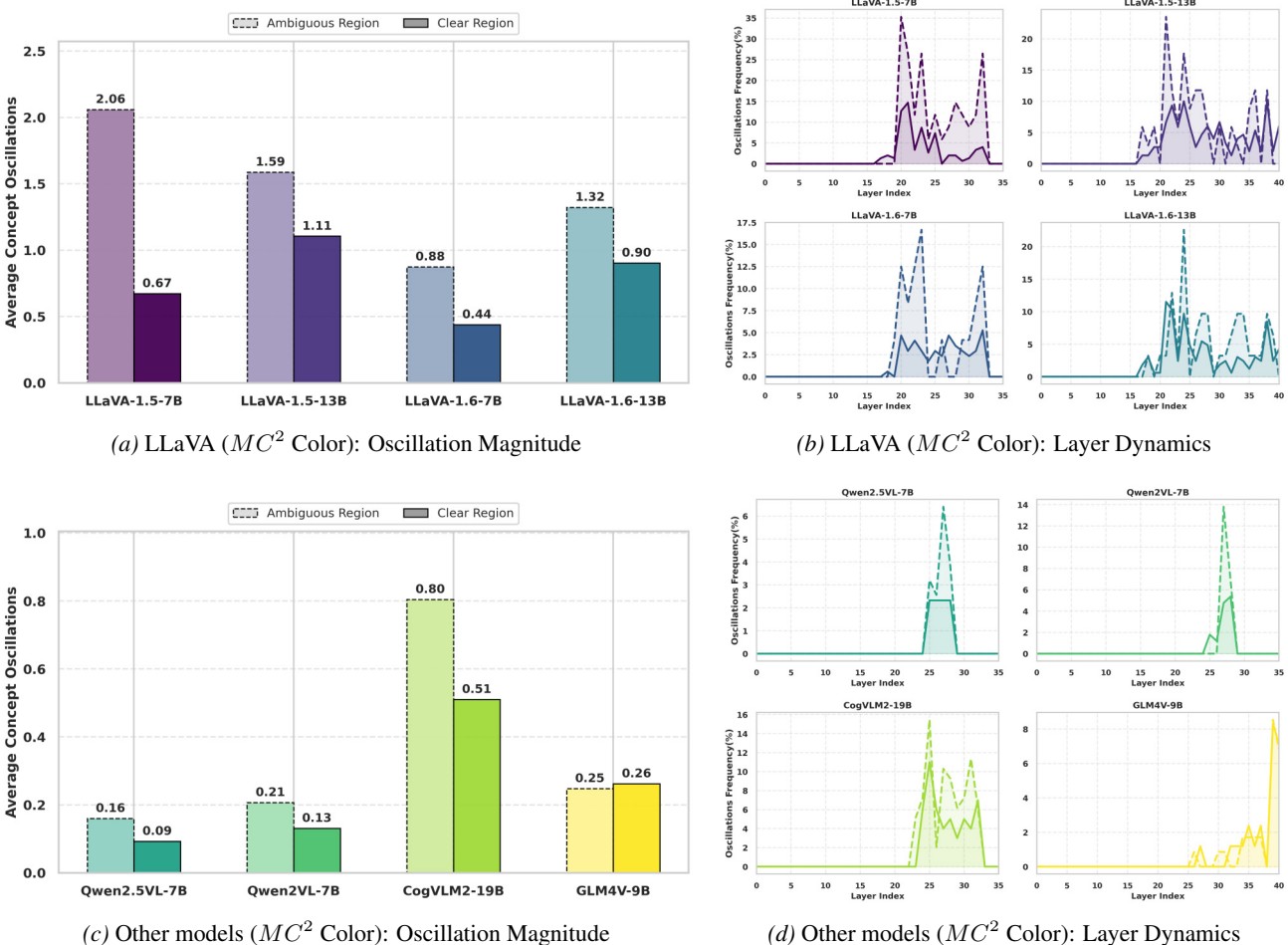

*(a)* LLaVA ($MC^2$ Color): Oscillation Magnitude

*(b)* LLaVA ($MC^2$ Color): Layer Dynamics

*(c)* Other models ($MC^2$ Color): Oscillation Magnitude

*(d)* Other models ($MC^2$ Color): Layer Dynamics

*Figure 17.* **Internal Oscillation Analysis on** $MC^2$ **Color Recognition.** The left column shows the statistical comparison of oscillation magnitude across different uncertainty regions. The right column visualizes the layer-wise evolution of logit differences. Note that the **Ambiguous** region consistently triggers the highest internal conflict.

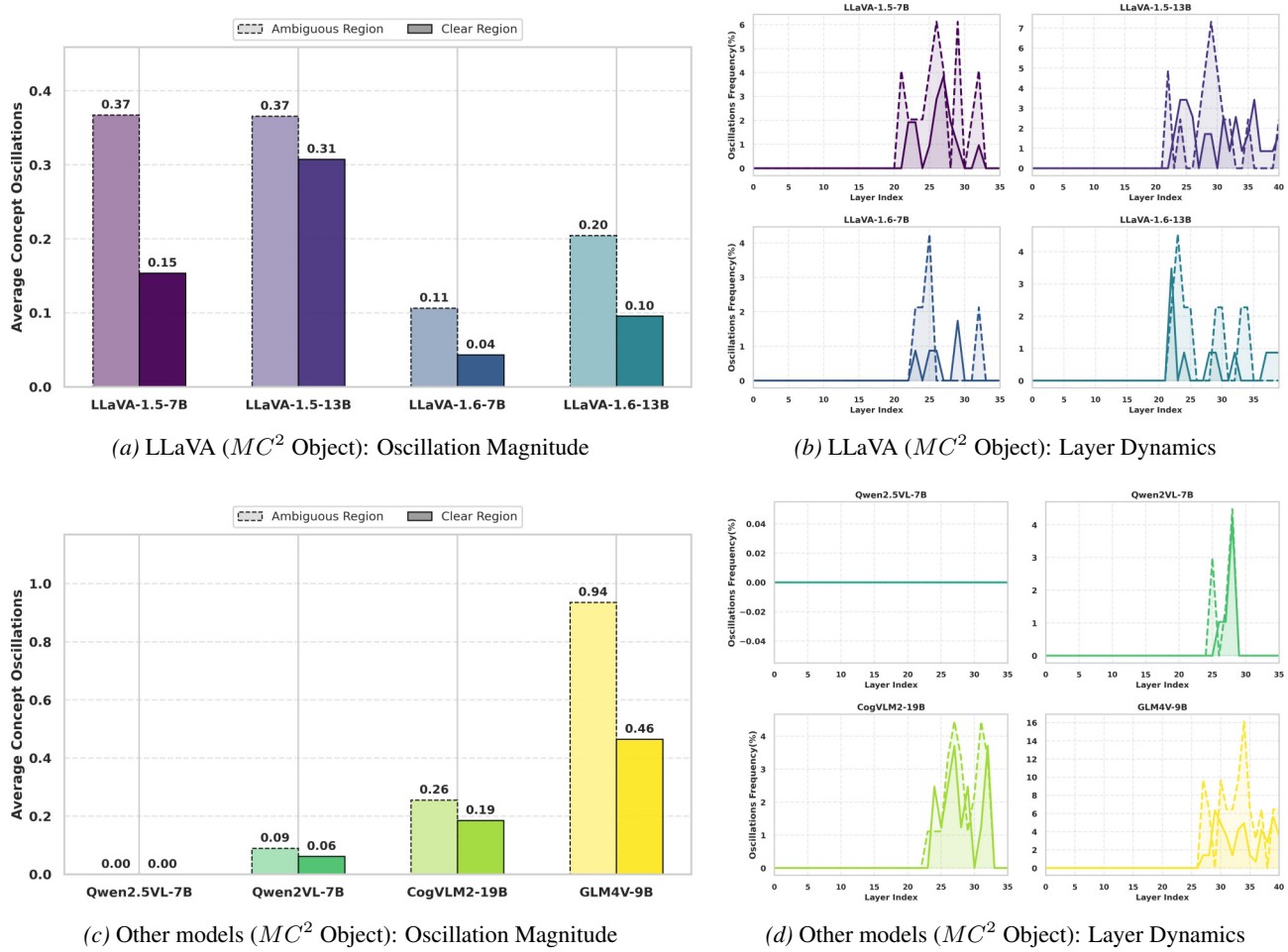

*(a)* LLaVA ($MC^2$ Object): Oscillation Magnitude

*(b)* LLaVA ($MC^2$ Object): Layer Dynamics

*(c)* Other models ($MC^2$ Object): Oscillation Magnitude

*(d)* Other models ($MC^2$ Object): Layer Dynamics

*Figure 18.* **Internal Oscillation Analysis on** $MC^2$ **Object Recognition.** Consistent with color tasks, object recognition tasks also exhibit significant internal oscillation in ambiguous regions, confirming the task-agnostic nature of this mechanism.

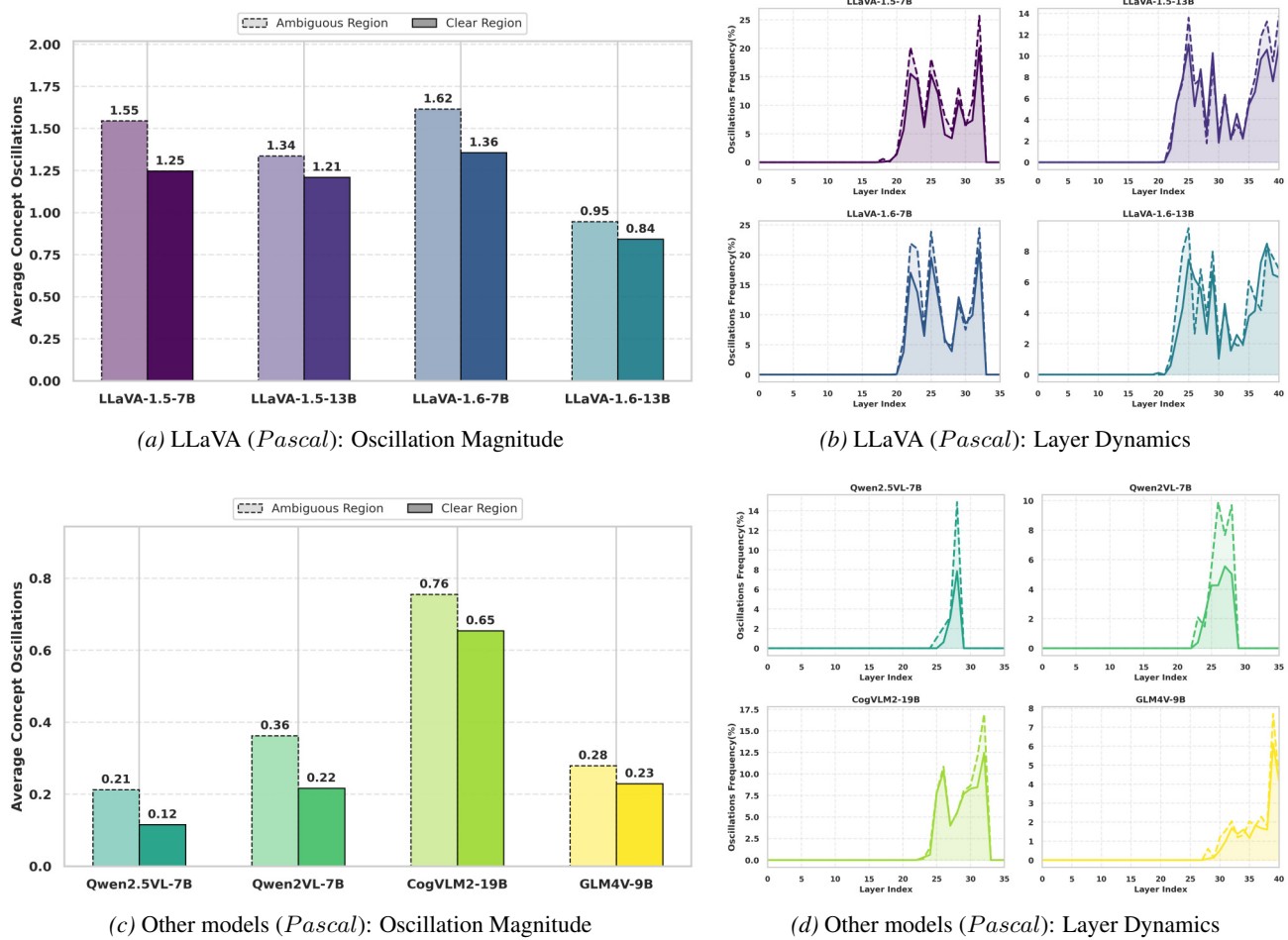

*(a)* LLaVA (*Pascal*): Oscillation Magnitude

*(b)* LLaVA (*Pascal*): Layer Dynamics

*(c)* Other models (*Pascal*): Oscillation Magnitude

*(d)* Other models (*Pascal*): Layer Dynamics

*Figure 19.* **Internal Oscillation Analysis on *Pascal* VOC Object Recognition.** Even in complex real-world datasets like Pascal VOC, the oscillation mechanism remains a robust indicator of multimodal conflict resolution.

## G. Spearman Rank Correlation Analysis: Per-Task Breakdown

Section 5 reports a prose summary of Spearman's rank correlation ($\rho$) between $\Delta H_{\mathrm{rel}}$ and the text-following probability across all 72 model-task pairs (8 models × 9 tasks). Here we provide the full per-model, per-task breakdown (Table 7), together with the rationale for using $\rho$ as the monotonicity measure and a discussion of the seven moderate-strength edge cases.

**Why Spearman's $\rho$.** Unlike Pearson correlation, Spearman's $\rho$ measures monotonicity without assuming a linear relationship between the two variables. Since the curve relating $\Delta H_{\mathrm{rel}}$ to text-following probability is typically a smooth sigmoid-like decrease (Figure 4a), a non-parametric measure such as $\rho$ is the appropriate statistical tool: it captures the strict monotonic ordering of probabilities along increasing $\Delta H_{\mathrm{rel}}$, regardless of the curve's exact shape.

**Per-Task Breakdown.** Table 7 reports $\rho$ for every one of the 72 model-task combinations (8 models × 9 tasks) evaluated in this paper. All 72 pairs yield negative $\rho$; 65/72 (~90.3%) fall below $-0.8$, and the vast majority reach $-0.95$ or lower.

**Edge Cases.** The seven pairs with $|\rho| < 0.8$ (highlighted by the absence of bold formatting in Table 7) fall into two categories. First, tasks whose empirical distribution of $\Delta H_{\mathrm{rel}}$ is concentrated in a narrow band (e.g., short binary-choice formats where the entropy gap is naturally compressed) reduce the statistical power for detecting strict ordering—this accounts for the moderate $\rho$ values on $MC^2$ color recognition for LLaVA-1.6-13B, Qwen2.5-VL-7B, and GLM-4V-9B, and on $MC^2$ sentiment understanding for LLaVA-1.6-7B and LLaVA-1.6-13B. Second, models saturated near one extreme

*Table 7.* Per-model, per-task Spearman rank correlation ($\rho$) between $\Delta H_{\rm rel}$ and text-following probability across all 72 model-task pairs. All values are negative; entries with $\rho < -0.8$ are shown in **bold** to indicate strong monotonic decrease.

| Dataset | Task | LLaVA-1.5-7B | LLaVA-1.5-13B | LLaVA-1.6-7B | LLaVA-1.6-13B | Qwen2-VL-7B | Qwen2.5VL-7B | CogVLM2-19B | GLM-4V-9B |
|---|---|---|---|---|---|---|---|---|---|
| MC$^2$ | activity | **-1.00** | **-0.95** | **-0.88** | **-0.98** | **-0.98** | **-0.98** | **-0.99** | **-0.96** |
| MC$^2$ | attribute | **-0.97** | **-0.90** | **-0.99** | **-1.00** | **-0.90** | **-0.92** | **-0.82** | **-0.88** |
| MC$^2$ | color | **-0.97** | **-0.89** | -0.77 | **-0.83** | **-0.83** | -0.55 | **-0.83** | -0.65 |
| MC$^2$ | object | **-0.86** | **-1.00** | **-1.00** | **-0.90** | **-0.90** | **-0.97** | **-0.96** | -0.57 |
| MC$^2$ | positional | **-0.98** | **-0.98** | **-0.98** | **-0.94** | **-0.95** | **-0.88** | **-0.99** | **-0.99** |
| MC$^2$ | sentiment | **-0.85** | **-0.83** | -0.76 | -0.79 | **-0.85** | **-0.98** | **-0.89** | **-0.98** |
| MC$^2$ | sport | **-0.97** | **-0.86** | **-1.00** | **-0.98** | **-0.99** | **-0.88** | **-0.99** | **-0.88** |
| Synthetic | color | **-1.00** | **-0.96** | **-1.00** | **-1.00** | **-1.00** | **-0.95** | **-0.99** | **-1.00** |
| Pascal VOC | color | **-0.94** | **-1.00** | **-1.00** | **-0.97** | **-0.99** | **-0.92** | **-1.00** | -0.37 |

of $\Delta H_{\rm rel}$ on a given task can exhibit weaker rank ordering, illustrated by GLM-4V-9B on Pascal VOC color recognition ($\rho = -0.37$) and on MC$^2$ object recognition ($\rho = -0.57$). Even in all these cases, the sign of $\rho$ remains negative, so the monotonic shape is preserved while the strength of the monotonic relationship is reduced. These outliers therefore highlight the importance of broad uncertainty coverage in our benchmark design rather than contradicting the main finding.

## H. DriveBench: Additional Experimental Details

This appendix complements Section 7.2 by providing further specifications of the DriveBench setup that were omitted from the main text for brevity.

**Benchmark Scope.** DriveBench is a multi-camera autonomous-driving benchmark covering object-centric perception (identifying nearby vehicles, pedestrians, lane elements), prediction (anticipating vehicle trajectories), and planning (selecting safe driving actions). We use the object-centric perception split for MCQ evaluation and a curated subset for long-form reasoning.

**Conflict Construction.** For each MCQ question, the ground-truth visual answer is paired with an automatically generated misleading text caption that names one of the non-ground-truth options. Conflict captions are generated via Gemini-3.1-Pro using a templated prompt that ensures the misleading caption is plausible (i.e., descriptive of an alternative scene element rather than syntactically broken text). All generated captions are filtered through human review for naturalness.

**Stratification Procedure.** Following the same protocol as the synthetic SFT experiment (recall $\Delta H_{\rm rel} = 2(H^{(t)} - H^{(v)})/(H^{(t)} + H^{(v)})$, so *lower* $\Delta H_{\rm rel}$ means text is more confident), training samples are partitioned by relative entropy:

- **Text-easier:** $\Delta H_{\rm rel} < -0.5$ — the (misleading) text appears more confident than the (correct) vision-only prediction.

- **Tie:** $|\Delta H_{\rm rel}| \leq 0.5$ — both modalities exhibit comparable uncertainty.

- **Vision-easier:** $\Delta H_{\rm rel} > 0.5$ — vision is more confident than the misleading text.

This matches the *Pos / Mid / Neg* convention of the main SFT experiment (Appendix E); all subsets are downsampled to the same size for fair comparison across conditions.

**MCQ Perception Results.** Table 8 reports the full MCQ perception results referenced from the main text (Section 7.2). The **Base** model collapses under conflict despite strong vision-only accuracy, and training on the **Text-easier** (boundary) subset consistently delivers the largest recovery for both models, well above the Tie and Vision-easier alternatives.

**Long-form Evaluation Protocol.** For open-ended reasoning, sequence-level entropy is computed as the average per-token entropy over the generated output. Conflict captions are produced with Gemini-3.1-Pro and the final answers are scored by Gemini-3-Flash, which acts as an automated judge rating coherence, factual grounding, and resistance to the misleading caption.

*Table 8.* **MCQ Perception on DriveBench (Conflict Input Setting).** Training on the **Text-easier** (boundary) subset yields the largest recovery from baseline collapse. "Vision-only Acc" shows the reference upper bound when no misleading caption is present.

| Model | Training subset | Conflict Acc % (↑) | Vision-only Acc % (↑) |
|---|---|---|---|
| LLaVA-1.5-7B | Base | 3.33 | 76.67 |
| | Text-easier | **50.00** | 83.33 |
| | Tie | 40.00 | 76.67 |
| | Vision-easier | 33.33 | 76.67 |
| Qwen2.5-VL-7B | Base | 6.67 | 83.33 |
| | Text-easier | **63.33** | 86.67 |
| | Tie | 53.33 | 83.33 |
| | Vision-easier | 43.33 | 80.00 |

**Discussion.** That the Text-easier subset—where the misleading caption appears *most* convincing—produces the largest robustness gain across both MCQ and long-form formats confirms the broader principle that data efficiency in preference alignment is governed by preference uncertainty. Training on these boundary cases teaches the model to override seemingly confident text with the visually grounded answer, and the gain transfers to a wide range of held-out conflict scenarios.

## I. Case Study: The Dynamics of Conflict.

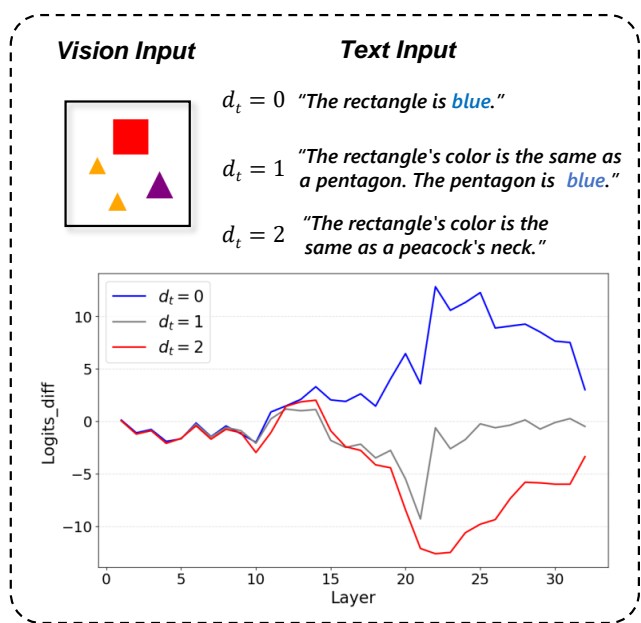

*Figure 20.* Case Study: Impact of Text Uncertainty on Layer-wise Confidence Dynamics.

We exemplify these dynamics using a concrete case on LLaVA-1.5-7B (Figure 20). By manipulating textual difficulty ($d_t = 0, 1, 2$) for a single image, we effectively shift the input across uncertainty regions. While easy and hard texts induce rapid, stable commitments to text (blue line) and vision (red line) respectively, the intermediate case ($d_t = 1$) falls into the ambiguous region. Here, the trajectory (gray line) vacillates near the zero-line decision boundary, visually capturing the internal hesitation driven by balanced uncertainty.

