# OpenReview forum: "Beyond Fixed Biases: Decoding the Role of Reasoning Uncertainty in MLLM Modality Conflicts"
_ICML.cc/2026/Conference — ICML 2026 regular_

### Official Review · Reviewer_5ENt · 2026-02-25

**Soundness:** 3
**Presentation:** 2
**Significance:** 4
**Originality:** 3
**Overall Recommendation:** 4
**Confidence:** 3

**Summary:**

This paper introduces the concept of "modality following" to systematically study how Multimodal Large Language Models resolve inter-modality conflicts, uncovering a universal law where the probability of following a modality decreases monotonically with its relative reasoning uncertainty. By analyzing this behavior, the authors identify a "balance point" that quantifies a model's inherent modality bias independent of unimodal capabilities, and reveal that conflict resolution involves high-level cognitive "concept oscillations" in the middle-to-late network layers. The framework further demonstrates practical utility for preference steering, showing that effective Supervised Fine-Tuning requires targeting boundary cases where modalities are in conflict, rather than easy samples dominated by a single modality.

**Compliance With Llm Reviewing Policy:**

Affirmed.

**Key Questions For Authors:**

1.Some figures do not show consistent trend for the same model. For example, in figure 3 (a), CogVLM2 reveals most preference ratio on text-following while in figure 3(b), CogVLM2 reveals equal distribution on relative uncertainty.

2.What is the boundary of ambiguous region and clear region for relative uncertainty?

3.What is the computation method of oscillations?

4.What is the application scenarios for preference steering in downstream tasks or real world model usage?

**Limitations:**

1.Please polish the language more carefully to increase the readability. The AI-style writing, especially the abstract and introduction is too hard to follow.

2.The reasoning ability is not the core focus in the paper since the study on modality following pattern is just one output word as shown in Figure 1, thus Reasoning Uncertainty in the title or abstract should be replaced into Preference Uncertainty to avoid misunderstanding.

3.Since the prediction accuracy would decreased largely following the increasing difficulty level, which leads to higher ratios of other part outcome except text/vision following outcome, this variance should be considered when comparing the modality preference in different difficulty levels.

4.More cases of both text and vision preference in various difficulty levels should be provided.

5.Find a better way to present the figure results to reduce readability difficulty. For example, change Pos/Neg/Mid in Figure 7 into Vision-easier/Text-easier/Tie legends descriptions.

**Strengths And Weaknesses:**

1.Preference bias in MLLM is worth being studied and self-constructed data in modality conflict situation is intuitive to realize this goal.

2.Visualization and experiements are plenty to lead to the corresponding conclusion.

3.Sigificance and soundness is good, while presentation is poor which require more language polish and visualization revision. Originality is acceptable.

---

> ### Author Rebuttal · Authors · 2026-03-29
>
> We deeply appreciate your recognition of both the significance and originality of our work.
>
> ## 1.1 On the Contradiction Between Final Behavior and Case Distribution:
> In **Figure 3(a)**, **CogVLM2** shows the highest preference for text-following, while in **Figure 3(b)**, it exhibits an even distribution of relative uncertainty. The former represents the **final behavior**, while the latter represents the **case distribution**. This apparent contradiction is precisely the motivation for our **decomposition**, which leads to the insight of the model’s inherent preference, the **Balance Point**. **CogVLM2**, with a highly visual-biased Balance Point, shows a dominant preference for text-following despite a more uniform distribution of uncertainty.
>
> ## 1.2 On Defining "Ambiguous" and "Clear" Regions:
> We will define the **“Ambiguous Region”** as the area around the Balance Point with a total width of 0.5, and the **“Clear Region”** as outside this range. In the vicinity of the Balance Point, the model’s modality-following behavior is closer to **equal probability**.
>
> ## 1.3 On Concept Oscillations:
> We define one oscillation as a switch in the model’s top-1 prediction between the text-supported answer and the image-supported answer across layers. For each sample, the total number of concept oscillations is the number of such switches observed during the forward pass.
>
> ## 1.4 On Preference Steering for Modalities:
> Preference steering can mitigate the  **modality bias** of MLLM in real-time video analysis or medical imaging diagnosis. By managing modality preferences effectively, we can prevent models from over-relying on a single modality. Besides, in domains that require more attention to a specific modality (e.g., autonomous driving), we can fine-tune the utilization of different information sources.
>
> ## 1.5 On Revisions for Clarity and Terminology:
> We have rephrased the abstract and introduction to improve readability and replaced "reasoning uncertainty" with "preference uncertainty" to avoid misunderstandings. Besides, We will add more examples of preference for text and vision in different difficulty levels in the appendix and improve the captions and annotations for all figures to reduce readability difficulty.
> Thank you for your valuable suggestions on presentation; all of these changes will be incorporated in the revised version.
>
> ## 1.6 More Experiments
> For additional task coverage and the quantitative monotonicity analysis, please also see our response to **Reviewer pBi5, Secs. 3.1–3.2**.

---

> > ### Author Rebuttal · Reviewer_5ENt · 2026-04-01
> >
> > Some limitations are not responded, including  reasoning ability, the modality preference comparison confusion in different difficulty levels.
> >
> > Besides, for Sec 1.4, why not use some particular domain like autonomous driving to further validate the effectiveness of the proposed method?

---

> > > ### Author Response · Authors · 2026-04-05
> > >
> > > Dear Reviewer 5ENt:
> > >
> > > We thank the reviewer for the constructive feedback. Below, we address each concern in detail and clarify the remaining points.
> > >
> > >
> > >
> > > **(1) On Reasoning vs. Preference**
> > >
> > > We agree that *reasoning ability* is not the primary focus of this work. Our goal is to study **how models resolve inter-modality conflicts at the decision level**, rather than evaluating reasoning quality itself.
> > >
> > > The “uncertainty” in our formulation is defined as a **relative signal within the model’s output space**, and should not be interpreted as a direct measure of reasoning capability. To avoid confusion, we will revise the terminology to **“preference uncertainty”** in the paper.
> > >
> > > Importantly, our findings are based on *controlled conflicting conditions*, and therefore reflect **modality selection behavior independent of absolute reasoning strength**.
> > >
> > > **(2) On Modality Preference Across Difficulty Levels**
> > >
> > > We appreciate the reviewer’s concern regarding potential confounding effects introduced by increasing difficulty.
> > >
> > > **(a) Conditioning on Valid Outcomes.**
> > > Our modality preference analysis is computed **conditioned on valid modality-following outcomes** (i.e., *text-following* or *vision-following*). Samples categorized as *other* (e.g., incorrect or ambiguous predictions) are excluded from the preference ratio computation. Therefore, the increase of “other” outcomes at higher difficulty levels does not bias our estimation of modality preference.
> > >
> > > **(b) Within-Level Consistency.**
> > > Our key conclusion is that the modality-following probability varies monotonically with relative uncertainty, and this relationship is observed within each difficulty level rather than relying on cross-level comparisons.
> > >
> > > **(c) Empirical Breakdown Across Difficulty Levels.**
> > > To further clarify this point, we provide a breakdown of outcome distributions across different difficulty settings (based on LLaVA-1.5 7B under the multimodal setting):
> > >
> > > | visual_difficulty | text_difficulty | total | visual (%)   | textual (%)  | other (%)   |
> > > | ----------------- | ----------------- | ----- | ------------ | ------------ | ----------- |
> > > | L1                | L1                | 400   | 147 (36.75%) | 250 (62.50%) | 3 (0.75%)   |
> > > | L1                | L2                | 400   | 185 (46.25%) | 215 (53.75%) | 0 (0.00%)   |
> > > | L1                | L3                | 400   | 381 (95.25%) | 13 (3.25%)   | 6 (1.50%)   |
> > > | L2                | L1                | 400   | 99 (24.75%)  | 250 (62.50%) | 51 (12.75%) |
> > > | L2                | L2                | 400   | 164 (41.00%) | 226 (56.50%) | 10 (2.50%)  |
> > > | L2                | L3                | 400   | 339 (84.75%) | 14 (3.50%)   | 47 (11.75%) |
> > >
> > > Importantly, our modality preference analysis is conducted on **valid samples only**, defined as cases where:
> > >  (i) both unimodal inputs (text-only and vision-only) lead to correct answers, and
> > >  (ii) the multimodal prediction clearly follows either the text-supported or vision-supported answer.
> > >
> > > Samples categorized as *other* (e.g., incorrect predictions or ambiguous outputs that do not follow either modality) are therefore excluded from the preference ratio computation. This ensures that our analysis focuses on **well-defined modality-following behavior**, rather than being confounded by general prediction failures.
> > >
> > > As shown above, while the proportion of *other* outcomes varies with difficulty, the **relative dominance between text-following and vision-following remains stable and interpretable within each setting**, supporting the validity of our preference analysis.
> > >
> > > Due to space constraints, we present only a subset of visual difficulty levels here; additional finer-grained stratifications, along with experiments using LLaVA-1.6 (7B/13B) and Qwen2.5-VL 7B under multimodal, text-only, and vision-only settings, will be included in the appendix.
> > >
> > > **(3) On Domain-Specific Validation (e.g., Autonomous Driving)**
> > >
> > > We thank the reviewer for this valuable suggestion.
> > >
> > > We are currently conducting additional experiments on **Drive-Bench**, a benchmark designed for autonomous driving scenarios, to further evaluate the effectiveness of our framework. Specifically, we randomly split a subset of samples as the training set, generate captions using Gemini 3.1 Pro, and construct Vision-easier/Text-easier/Tie legends partitions (following the reviewer’s suggestion) based on our proposed difficulty stratification. We then fine-tune the model on these curated samples and evaluate its performance under multimodal settings.
> > >
> > > **For more details, please also refer to the first part of our reply to Reviewer HpY4, where we provide a more complete description of this additional validation.**
> > >
> > > We believe such domain-specific validation is complementary to our current contribution and further demonstrates the practical applicability of our framework.
> > >
> > > We hope that our clarifications adequately address your concerns and would appreciate your reconsideration of the rating.

---

### Official Review · Reviewer_pBi5 · 2026-03-09

**Soundness:** 3
**Presentation:** 3
**Significance:** 3
**Originality:** 4
**Overall Recommendation:** 4
**Confidence:** 4

**Summary:**

The paper investigates how multimodal LLMs resolve conflicts between vision and text.
The authors decompose this behavior into case-specific reasoning uncertainty and the model’s stable inherent preference.
Across multiple MLLMs and three datasets, they observe that (1) the probability of following a modality decreases as its relative uncertainty increases (2) concept oscillations” localized to middle-to-late layers in ambiguous cases, and (3) SFT for preference steering is more effective when training on “boundary” cases with balanced uncertainties than on “easy” cases.

**Compliance With Llm Reviewing Policy:**

Affirmed.

**Final Justification:**

The rebuttal addressed my concerns.

**Key Questions For Authors:**

1. The controlled experiments mainly involve perceptual tasks (e.g., color or object recognition); do the authors expect the proposed uncertainty law and balance-point interpretation to hold in more complex multimodal reasoning settings such as spatial reasoning, diagram reasoning, or multi-step visual reasoning?

2. The paper interprets layer-wise modality oscillations as evidence of conflict resolution dynamics; have the authors considered intervention-based analyses (e.g., layer frozen alignment) to determine whether these oscillations causally influence the final modality-following decision?

3. Could you adjust the page's graphs to pdf so that some figure captions and axis labels could be clearer. The fonts of most plots are hard to read at normal scales

**Limitations:**

The discussion of methodological limitations is absent. Some weakness could be covered in weakness.

**Strengths And Weaknesses:**

**Strength**

*Soundness:* The paper proposes a well-motivated analytical framework that decomposes modality-following behavior into relative reasoning uncertainty and inherent modality preference, supported by consistent empirical evidence across multiple models and datasets.

*Presentation:* The paper is clearly structured, with well-defined concepts (e.g., relative uncertainty and balance point) and informative visualizations that effectively illustrate the proposed analysis.

*Significance:* The work addresses an important and under-explored problem in multimodal reasoning: how MLLMs resolve conflicting information across modalities. It provides insights that may inform both evaluation and training strategies.

*Originality:* The paper introduces a novel perspective of multimodal model interpretability.

**Weakness**

1. The evaluation set might be limited. The paper's whole experiments are based on relatively simple problems with very short answers (colors, object categories, shapes, materials). There is no evaluation on more open-ended VQA, caption editing, or instruction-following scenarios where multimodal reasoning is more complex. This restricts the external validity of the **law** and of the **boundary cases are essential** conclusion.

2. The claim might be overstated. The paper repeatedly refers to a “fundamental law” or “universal monotonic law” governing modality following, yet the evidence might be limited. There is no quantitative measure of monotonicity (e.g., monotone regression fit, monotonicity violation rate) nor any formal statistical testing.

3. The analysis focuses exclusively on conflicting image-text inputs without including matched-modality control cases (e.g., image and text providing the same answer). Such matched settings could serve as an important baseline to disentangle modality conflict from inherent task difficulty. For instance, if a task is intrinsically challenging for the model, high uncertainty or incorrect predictions may arise even when the two modalities agree. Without this control condition, it is difficult to determine whether the observed uncertainty patterns are driven by modality arbitration or simply by task difficulty.

---

> ### Author Rebuttal · Authors · 2026-03-29
>
> We would like to sincerely thank you for recognizing the importance of our research question.
>
> ## 1.1 On Task Complexity and Generalizability:
> We fully recognize that evaluating more complex task settings would further strengthen the external validity of our findings. **Beyond the original settings, now we further expand the evaluation to additional MC² tasks, including activity recognition, attribute, sentiment understanding, sport recognition, and positional reasoning. And the observed patterns remain consistent across these diverse task types.** The results are in https://anonymous.4open.science/r/rebuttal-47EBAFABKFB
> We will include these additional results in the revised appendix.
>
>
> ## 1.2 On Statistical Analysis and Monotonicity:
> We computed Spearman rank correlations between modality-following ratio and relative uncertainty across models and tasks. Please see our response to **Reviewer 2wKV, Sec. 1.2**. Across all 72 model-task pairs, all coefficients are negative, and more than 90% are below -0.8. This provides direct quantitative support for a strong monotonic relationship. In the revision, we will report these statistics alongside the plots. We also agree that the wording should remain measured, and we are happy to phrase the claim as a consistent monotonic trend or empirical regularity rather than an overstated universal law.
>
>
> ## 1.3 On Matched-Modality Controls and Task Difficulty
> We agree that matched-modality controls are important for disentangling conflict effects from inherent task difficulty. We therefore added a matched-modality control in our intervention study, where image and text support the same answer while keeping the underlying task instance comparable. As reported below in **Sec. 1.4**, the average oscillation count in this agreement setting is only 19% of that under modality conflict, suggesting that the large oscillation pattern is not simply due to task difficulty alone. More broadly, our claim is not that uncertainty is produced only by conflict; rather, both unimodal capability and task difficulty contribute to uncertainty, and modality-following behavior depends on their relative difference.
>
> ## 1.4 On Intervention-Based Analysis
> We conducted an additional control and intervention study on the synthetic color-recognition dataset using LLaVA-1.5-7B. For samples in the ambiguous region, we first measured oscillations under a matched-modality control condition, where image and text support the same answer. The average oscillation count in this agreement setting is only 19% of that under modality conflict, suggesting that the baseline level of fluctuation without cross-modal disagreement is substantially lower.
>
> We further performed an activation-patching intervention in later layers to examine whether oscillations are functionally related to the final modality-following decision. Specifically, for layers where the model’s top-1 prediction switches to the answer opposite to the target modality, we replaced the hidden states with those from the matched-modality condition. Patching toward the text-supported answer leads to 94% text following while reducing the average oscillation count by 73%; patching toward the image-supported answer leads to 95% vision following while reducing the average oscillation count by 60%. These results suggest that suppressing oscillations substantially shifts the final modality preference toward the patched modality.
>
> We repeated the same analysis on LLaVA-1.6-7B and observed similar trends. Together, these findings provide additional support that the oscillations we analyze are closely associated with conflict-resolution dynamics, while the matched-modality condition offers a baseline for fluctuation unrelated to modality disagreement.
>
> ## 1.5 On Improving Figure Readability
> We will replace the raster figures with PDF/vector versions in the revised paper to improve the readability of captions, legends, and axis labels.

---

> > ### Author Rebuttal · Reviewer_pBi5 · 2026-04-03
> >
> > Follow-up questions regarding to:
> >
> > 1.1 Are there any rational to select MC^2 as extended evaluation?

---

> > > ### Author Response · Authors · 2026-04-04
> > >
> > > Thank you for your reply.
> > >
> > > We select MC² as an extended evaluation benchmark for three main reasons:
> > > 1. **Realism beyond synthetic settings**
> > >
> > >     Compared with our controllable synthetic data, MC² is constructed from real images paired with conflicting textual contexts, and its answers are human-validated for each modality. This makes it a substantially less controlled but more realistic testbed for multimodal conflict resolution. Evaluating on MC² allows us to examine whether the proposed uncertainty-following law and balance-point interpretation generalize beyond synthetic constructions.
> > >
> > > 2. **Broader task coverage**
> > >
> > >     Unlike our diagnostic data with controlled templates, MC² is derived from TDIUC and covers multiple perceptual reasoning tasks. In our experiments, we focus on its color and object-recognition subsets (filtered to single-word answers for clean evaluation). This diversity introduces more heterogeneous conflict patterns, providing a stronger test of the external validity of our framework.
> > >
> > > 3. **Community adoption**
> > >
> > >     MC² has been adopted in subsequent studies on modality preference and arbitration (e.g., [1,2,3]), making it a relevant and recognized benchmark for evaluating multimodal conflict behavior.
> > >
> > >
> > > Overall, MC² complements our controlled benchmark by trading off controllability for realism and diversity, enabling a more comprehensive evaluation of our framework. We hope this clarifies the rationale and addresses your concern.
> > >
> > > ---
> > >
> > > [1] Huang Z, Li X, Surana R, et al. AMPS: Adaptive Modality Preference Steering via Functional Entropy[J]. arXiv preprint arXiv:2602.12533, 2026.
> > >
> > > [2] Zhang Y, Xu M, Bai X, et al. Instruction Anchors: Dissecting the Causal Dynamics of Modality Arbitration[J]. arXiv preprint arXiv:2602.03677, 2026.
> > >
> > > [3] Pang B. Investigating stereotypical bias in large language and vision-language models[J]. Dissertation, University of Auckland, New Zealand, 2025.

---

### Official Review · Reviewer_HpY4 · 2026-03-12

**Soundness:** 2
**Presentation:** 2
**Significance:** 3
**Originality:** 2
**Overall Recommendation:** 4
**Confidence:** 5

**Summary:**

This paper studies how multimodal large language models resolve conflicts between modalities when image and text provide contradictory information. The authors propose a framework that decomposes modality-following behavior into relative reasoning uncertainty (estimated via unimodal entropy) and inherent modality preference. Empirically, the paper reports a monotonic relationship between relative uncertainty and the probability of following a modality, allowing the definition of a balance point that quantifies intrinsic preference independent of unimodal capability. The authors further analyze internal model dynamics via layer-wise probing and observe concept oscillations in middle-to-late layers when the model encounters ambiguous modality conflicts. Finally, the framework is applied to supervised fine-tuning, showing that training on boundary cases with high uncertainty improves preference steering more effectively than training on easy samples.

**Compliance With Llm Reviewing Policy:**

Affirmed.

**Final Justification:**

Thank the authors for the rebuttal.

**Key Questions For Authors:**

See Weaknesses Above.

**Strengths And Weaknesses:**

Strengths:
1. The decomposition of modality-following behavior into relative uncertainty and inherent preference provides a useful lens for analyzing modality bias in MLLMs.
2. The synthetic datasets allow independent manipulation of visual and textual difficulty, enabling more precise analysis of modality conflict scenarios.

Weaknesses:
1. The central claim is: "The probability of following a modality decreases as its uncertainty increases.". A probabilistic model naturally behaves like: $P(\text{modality}) \propto \text{confidence}$. So the monotonic relationship between confidence difference and decision probability is not surprising. The main empirical observation appears intuitive and may arise naturally from probabilistic decoding.
2. The paper measures uncertainty using token entropy. I am concerned whether this is simply comparable across modalities, for example, vision tokens and text tokens might have different distributions, calibration differs between models. This not reflect true reasoning uncertainty.
3. Experiment-wisely, the datasets focus on simple recognition tasks and may not capture the complexity of real multimodal reasoning.
4. Some related works can be discussed:

[1] Han, Junlin, Huangying Zhan, Jie Hong, Pengfei Fang, Hongdong Li, Lars Petersson, and Ian Reid. "What Images are More Memorable to Machines?." arXiv preprint arXiv:2211.07625 (2022).

[2] Endo, Mark, and Serena Yeung-Levy. "Downscaling Intelligence: Exploring Perception and Reasoning Bottlenecks in Small Multimodal Models." arXiv preprint arXiv:2511.17487 (2025).

---

> ### Author Rebuttal · Authors · 2026-03-29
>
> Thank you very much for your positive feedback on the analytical framework we have proposed.
>
> ## 1.1 On the Intuition of Probability and Confidence:
> Our contribution is not to deny this intuition, but to make it measurable in controlled multimodal conflict settings and to separate two components that are conflated in aggregate statistics: case-specific relative uncertainty and model-specific inherent preference. Concretely, we quantify how modality-following probability changes with relative uncertainty, introduce the balance point as a measure of inherent preference, and analyze layer-wise oscillations around this balance point. In this sense, our contribution is a controlled decomposition and mechanistic analysis of conflict resolution, rather than the qualitative claim that “confidence should matter.” This is a part that other papers have attempted to analyze but lack in-depth exploration, such as in [1].
>
> [1] Words or Vision: Do Vision-Language Models Have Blind Faith in Text?
>
> ## 1.2 On Entropy and Cross-Modality Comparisons:
> We use the **relative uncertainty** metric ($\Delta H_{rel}$), which mitigates the issues of dimensionality in direct cross-modality comparisons. Entropy serves as a **proxy measure** in this context, and the core method focuses on comparing answer tokens, ensuring that the answer space between modalities is aligned. We limit the entropy and uncertainty comparison to the same model to ensure robustness. In **Appendix F.4**, we demonstrate how the trend holds across different metrics, such as **Negative Log-Likelihood (NLL)** and **Margin**, proving that the underlying property is independent of the metric used, and not a characteristic of entropy itself.
>
> ## 1.3 On Task Complexity and Generalizability:
> We fully recognize that evaluating more complex task settings would further strengthen the external validity of our findings. **Beyond the original settings, now we further expand the evaluation to additional MC² tasks, including activity recognition, attribute, sentiment understanding, sport recognition, and positional reasoning. And the observed patterns remain consistent across these diverse task types.** The results are in https://anonymous.4open.science/r/rebuttal-47EBAFABKFB
>
> ## 1.4 On Related Work:
> The article you referenced is invaluable for positioning our work within the current field. We will incorporate a discussion of this work in the related work section of the revised version.

---

> > ### Author Rebuttal · Reviewer_HpY4 · 2026-04-06
> >
> > Thank you for the rebuttal. However, I think the paper/rebuttal requires further justifications.
> >
> > 1. The practical value remains unclear. The evaluation is still limited in scope, and it is not evident whether the proposed framework leads to meaningful improvements in realistic multimodal settings. This makes it difficult to assess its significance beyond the current experimental setup. Can the authors show insights around CoT generation as for realistic (no-toy) setting.
> >
> > 2. The rebuttal largely rephrases the original contributions rather than addressing the core concern of novelty. It does not sufficiently explain why the observed behavior is non-trivial or cannot be explained by simpler confidence-based mechanisms. I hope the authors can clarify this problem.
> >
> > I remain open to raising my score if the authors can address these critical concerns with convincing experiments and stronger justification.

---

> > > ### Author Response · Authors · 2026-04-07
> > >
> > > > 1. The practical value remains unclear.
> > > ## Response
> > >
> > > We thank the reviewer for raising the concern about realistic validation. To address this, we add a new experiment on **DriveBench[1]**, a real-world multi-camera autonomous-driving benchmark covering **object-centric perception**, **prediction**, and **planning**.
> > >
> > > ## Experimental Setup
> > >
> > > - **MCQ perception.** For each object-centric perception question, we construct **answer-targeted conflict captions** for all non-ground-truth options(non-vision-supported options). For example, if the correct answer is **A**, we generate separate conflict captions supporting **B**, **C**, and **D**. Each training sample is formed as **input** (correct image + wrong caption + question) and **target** (the visually correct answer). This creates targeted cross-modal conflicts, rather than arbitrary text perturbations.
> > >
> > > - **Entropy-based partition.** We split the data into train/test sets and compute the relative entropy. Training examples are then partitioned into three subsets:
> > >   - **Text-easier**: the misleading text is more confident than vision;
> > >   - **Tie**: text and vision have similar uncertainty;
> > >   - **Vision-easier**: vision is more confident than the misleading text.
> > >
> > >   All subsets are **budget-matched** and **balanced** over both the correct answer and the targeted wrong option.
> > >
> > >      - **MCQ evaluation metrics.** We report:
> > >           - **Conflict Accuracy**: accuracy under contradictory multimodal input; this is the main metric, measuring whether the model can resist misleading text and follow visual evidence.
> > >           - **Vision-only Accuracy**: accuracy with image + question only; this reflects the model’s underlying visual capability on the task.
> > >
> > > - **Long-form planning and prediction.** We further extend the setting to long-form **planning** and **prediction** tasks. For each example, we generate a plausible but incorrect target answer and a supporting conflict caption with **Gemini-3.1-Pro**, while keeping the supervision target as the official ground-truth answer.
> > >
> > >     - **Long-form entropy and evaluation.** For long-form reasoning, we compute sequence-level entropy as the **average token entropy over the output sequence**: $H_{\text{seq}}=\frac{1}{L}\sum_{i=1}^{L} H_i.$
> > >   We compare the average token entropy of the correct answer under the **vision-only** condition with that of the text-supported wrong answer under the **text-only** condition, and apply the same **Text-easier / Tie / Vision-easier** partition. For evaluation, we use **Gemini-2.5-Flash** as an automatic judge to compare the model output with the official reference answer. The reported **Overall Score** measures the overall quality and correctness of the generated long-form response.
> > >
> > > ## Experimental Results
> > >
> > > We run both **LLaVA-1.5-7B** and **Qwen2.5-VL-7B** under the same scene-disjoint split. As shown in Tables 1 and 2, across both MCQ perception and long-form reasoning, conflict supervision consistently reduces misleading-text following, and **Text-easier** samples provide the strongest visual-correction effect. This suggests that our framework extends beyond the original controlled setup to realistic autonomous-driving perception and long-form multimodal reasoning.
> > >
> > > #### Table 1. MCQ Perception Results  in Conflict Input Setting on DriveBench
> > > | Model | Subset | Conflict Acc %($\uparrow$) | Vision-only Acc% ($\uparrow$) |
> > > |---|---:|---:|---:|
> > > | LLaVA-1.5-7B | Base | 3.33 | 76.67 |
> > > | LLaVA-1.5-7B | Text-easier | **50.00** | 83.33 |
> > > | LLaVA-1.5-7B | Tie | 40.00 | 76.67 |
> > > | LLaVA-1.5-7B | Vision-easier | 33.33 | 76.67 |
> > > | Qwen2.5-VL-7B | Base | 6.67 | 83.33 |
> > > | Qwen2.5-VL-7B | Text-easier | **63.33** | 86.67 |
> > > | Qwen2.5-VL-7B | Tie | 53.33 | 83.33 |
> > > | Qwen2.5-VL-7B | Vision-easier | 43.33 | 80.00 |
> > >
> > > #### Table 2. Reasoning Results in Conflict Input Setting on DriveBench
> > > | Model | Subset | Conflict Overall Score($\uparrow$) |
> > > |---|---:|---:|
> > > | LLaVA-1.5-7B | Base | 13.93 |
> > > | LLaVA-1.5-7B | Text-easier | **22.88** |
> > > | LLaVA-1.5-7B | Tie | 20.10 |
> > > | LLaVA-1.5-7B | Vision-easier | 18.25 |
> > > | Qwen2.5-VL-7B | Base | 32.81 |
> > > | Qwen2.5-VL-7B | Text-easier | **45.35** |
> > > | Qwen2.5-VL-7B | Tie | 37.32 |
> > > | Qwen2.5-VL-7B | Vision-easier | 34.64 |
> > > > 2. Why the observed behavior is non-trivial
> > >
> > > While a simple confidence-based account can explain the general tendency that models prefer the less uncertain modality, it cannot explain why different models exhibit systematically different following behaviors under similar relative-uncertainty conditions. Our novelty is therefore not the monotonicity itself, but the decomposition of modality following into case-specific relative uncertainty and model-specific inherent preference, quantified by the balance point, together with its mechanistic and training implications.
> > >
> > > [1] Are VLMs Ready for Autonomous Driving? An Empirical Study from the Reliability, Proceedings of the IEEE/CVF International Conference on Computer Vision. 2025: 6585-6597.

---

### Official Review · Reviewer_2wKV · 2026-03-13

**Soundness:** 2
**Presentation:** 3
**Significance:** 3
**Originality:** 2
**Overall Recommendation:** 4
**Confidence:** 2

**Summary:**

This paper studies how multimodal large language models resolve conflicts between image and text inputs, a behavior the authors call modality following. The main proposal is a decomposition of this behavior into case-specific relative reasoning uncertainty, estimated from unimodal output entropy, and a model-specific inherent modality preference, quantified by a balance point where the model is equally likely to follow either modality. Empirically, across synthetic and real-world conflict datasets, the paper reports a monotonic relationship between relative uncertainty and modality-following probability, analyzes internal layer-wise oscillations near the balance point, and uses this framework to guide data selection for supervised fine-tuning aimed at steering modality preference.

**Compliance With Llm Reviewing Policy:**

Affirmed.

**Final Justification:**

This paper studies how multimodal large language models resolve conflicts between image and text inputs. The rebuttal resolved my concerns and I will keep my score.

**Key Questions For Authors:**

See Weaknesses.

**Limitations:**

See Weaknesses.

**Strengths And Weaknesses:**

Strengths:
1. This paper tackles an important question for MLLMs, namely, how they arbitrate between conflicting modalities. And the primary empirical observation is intuitive and potentially useful.
2. The paper uses a reasonably broad set of models and combines controllable synthetic data with two real-world benchmarks.
3. The analysis of this paper is interesting, and the preference-steering application is practically motivated.

Weaknesses:
1. The paper relies heavily on visual trends without enough statistical reporting, and it's better to use table-style quantitative reporting.
2. It's better for the authors to provide stronger baselines for data selection and clearer quantitative reporting.

---

> ### Author Rebuttal · Authors · 2026-03-29
>
> First, thanks for recognizing the importance of the problem we are addressing.
>
> ## 1.1 More Experiments
> Beyond the original settings, we have now expanded the evaluation to additional MC² tasks, including activity recognition, attribute, sentiment understanding, sport recognition, and positional reasoning. The same monotonic trend remains consistent across these tasks, suggesting that our observations are not limited to the originally reported color/object settings. Please also see our response to **Reviewer pBi5, Sec. 1.1**, where we summarize these newly added experiments.
>
> ## 1.2 Regarding the request for more quantitative data in table form:
> We have computed the **Spearman’s Rank Correlation** matrix for various models across different datasets:
> | Benchmark | Task | LLaVA-1.5-13B | LLaVA-1.5-7B | LLaVA-1.6-13B | LLaVA-1.6-7B | Qwen2VL-7B | Qwen2.5VL-7B | CogVLM2-19B | GLM4V-9B |
> |---|---|---:|---:|---:|---:|---:|---:|---:|---:|
> | MC^2 | activity recognition | -0.9524 | -1.0000 | -0.9833 | -0.8810 | -0.9833 | -0.9762 | -0.9940 | -0.9624 |
> | MC^2 | attribute | -0.9048 | -0.9701 | -1.0000 | -0.9916 | -0.8982 | -0.9222 | -0.8193 | -0.8833 |
> | MC^2 | color recognition | -0.8857 | -0.9701 | -0.8286 | -0.7731 | -0.8333 | -0.5543 | -0.8333 | -0.6467 |
> | MC^2 | object recognition | -1.0000 | -0.8571 | -0.9000 | -1.0000 | -0.9000 | -0.9701 | -0.9643 | -0.5663 |
> | MC^2 | positional reasoning | -0.9818 | -0.9818 | -0.9364 | -0.9818 | -0.9537 | -0.8838 | -0.9909 | -0.9886 |
> | MC^2 | sentiment understanding | -0.8333 | -0.8503 | -0.7904 | -0.7619 | -0.8500 | -0.9762 | -0.8870 | -0.9762 |
> | MC^2 | sport recognition | -0.8571 | -0.9667 | -0.9762 | -0.9958 | -0.9940 | -0.8815 | -0.9940 | -0.8810 |
> | Synthetic | color recognition | -0.9643 | -1.0000 | -1.0000 | -1.0000 | -1.0000 | -0.9515 | -0.9910 | -1.0000 |
> | Pascal VOC | color recognition | -0.9958 | -0.9429 | -0.9662 | -1.0000 | -0.9910 | -0.9175 | -1.0000 | -0.3660 |
>
> Across all 72 pairs, the coefficients are consistently negative, and more than 90% of them are below -0.8, indicating a robust monotonic relationship between the modality-following ratio and relative uncertainty across models and tasks.

---

> > ### Author Rebuttal · Reviewer_2wKV · 2026-04-04
> >
> > Thanks for the rebuttal, I will keep my score.

---

### Decision · Program_Chairs · 2026-04-30

**Decision:**

Accept (regular)

**Comment:**

In this paper, the authors propose a framework that decomposes modality-following behavior into relative reasoning uncertainty (estimated via unimodal entropy) and inherent modality preference. Originally, reviewers have concerns such as insufficient statistical reporting, overstated claims, etc. After the rebuttal, all reviewers said that their concerns were well resolved by the authors, and recommended acceptance of the paper. The AC carefully read the paper, the rebuttal, and the reviewer discussions, and think the paper has good contribution to the community; and thus recommends acceptance of the paper.